# Key steps in unconventional secretion of fibroblast growth factor 2 reconstituted with purified components

Julia P Steringer[1], Sascha Lange[2†§], Sabína Čujová[3†], Radek Šachl[3†], Chetan Poojari[4,5†], Fabio Lolicato[4,5†], Oliver Beutel[6†], Hans-Michael Müller[1], Sebastian Unger[1], Ünal Coskun[7,8‡], Alf Honigmann[6‡], Ilpo Vattulainen[4,5,9‡], Martin Hof[3‡], Christian Freund[2‡], Walter Nickel[1*]

[1]Heidelberg University Biochemistry Center, Heidelberg, Germany; [2]Institut für Chemie und Biochemie, Freie Universität Berlin, Berlin, Germany; [3]J. Heyrovský Institute of Physical Chemistry, Academy of Sciences of the Czech Republic, Prague, Czech Republic; [4]Department of Physics, University of Helsinki, Helsinki, Finland; [5]Department of Physics, Tampere University of Technology, Tampere, Finland; [6]Max Planck Institute of Molecular Cell Biology and Genetics, Dresden, Germany; [7]Paul Langerhans Institute Dresden, Helmholtz Zentrum München, University Hospital and Faculty of Medicine Carl Gustav Carus, Technische Universität Dresden, Dresden, Germany; [8]Deutsches Zentrum fur Diabetesforschung, Neuherberg, Germany; [9]MEMPHYS – Center for Biomembrane Physics, University of Southern Denmark, Denmark, United Kingdom

*For correspondence: walter. nickel@bzh.uni-heidelberg.de

[†]These authors contributed equally to this work
[‡]These authors also contributed equally to this work

Present address: [§]Department of Molecular Biophysics, Leibniz-Forschungsinstitut fürMolekulare Pharmakologie, Berlin, Germany

Competing interests: The authors declare that no competing interests exist.

**Abstract** FGF2 is secreted from cells by an unconventional secretory pathway. This process is mediated by direct translocation across the plasma membrane. Here, we define the minimal molecular machinery required for FGF2 membrane translocation in a fully reconstituted inside-out vesicle system. FGF2 membrane translocation is thermodynamically driven by $PI(4,5)P_2$-induced membrane insertion of FGF2 oligomers. The latter serve as dynamic translocation intermediates of FGF2 with a subunit number in the range of 8-12 FGF2 molecules. Vectorial translocation of FGF2 across the membrane is governed by sequential and mutually exclusive interactions with $PI(4,5)P_2$ and heparan sulfates on opposing sides of the membrane. Based on atomistic molecular dynamics simulations, we propose a mechanism that drives $PI(4,5)P_2$ dependent oligomerization of FGF2. Our combined findings establish a novel type of self-sustained protein translocation across membranes revealing the molecular basis of the unconventional secretory pathway of FGF2.

## Introduction

Fibroblast Growth Factors (FGFs) form a family of more than 20 potent mitogens that stimulate the growth of a wide range of cells including fibroblasts and endothelial cells (*Bikfalvi et al., 1997*). They are of critical importance for physiological processes such as embryonic development, tissue regeneration, wound repair and hematopoiesis (*Bikfalvi et al., 1997*). FGF2 is the prototype member of this family that, beyond the functions of FGFs in normal cell growth and differentiation, plays critical roles under pathophysiological conditions (*Akl et al., 2016*). This is particularly evident in the context of cancer with FGF2 being a major mediator of tumor-induced angiogenesis (*Presta et al., 2005*). In addition, FGF2 acts as a survival factor that inhibits tumor cell apoptosis by an autocrine

secretion-signaling loop (*Noh et al., 2014*; *Pardo et al., 2006*). This process is believed to represent a frequent cause of tumor cell resistance against conventional anti-cancer therapies.

As opposed to other FGF family members, FGF2 lacks a signal peptide and is transported into the extracellular space by an ER/Golgi-independent mechanism (*La Venuta et al., 2015*; *Nickel, 2011*; *Nickel and Rabouille, 2009*, *2008*). The term unconventional secretion has been used for four different pathways of protein transport towards the plasma membrane and the extracellular space (*Rabouille et al., 2012*). Two of these pathways concern soluble proteins derived from the cytoplasm that are transported into the extracellular space by either direct protein translocation across the plasma membrane (type I unconventional secretion) or vesicular mechanisms involving endosomal compartments (type III unconventional secretion) (*Rabouille et al., 2012*; *Zhang and Schekman, 2013*; *Piccioli and Rubartelli, 2013*). Based on compelling evidence from both cell-based FGF2 secretion experiments and biochemical in vitro studies, unconventional secretion of FGF2 was shown to follow a type I mechanism that is based on direct protein translocation across the plasma membrane (*La Venuta et al., 2015*; *Zhang and Schekman, 2013*; *Malhotra, 2013*; *Nickel, 2005*; *Schäfer et al., 2004*). This process depends on sequential interactions of FGF2 with the phosphoinositide $PI(4,5)P_2$ at the inner leaflet and heparan sulfate proteoglycans at the outer leaflet of the plasma membrane (*Nickel, 2011*; *Nickel and Rabouille, 2009*; *Nickel, 2007*; *Temmerman et al., 2008*; *Zehe et al., 2006*). Recruitment by $PI(4,5)P_2$ triggers oligomerization of FGF2 at the inner leaflet resulting in membrane insertion, a key step in FGF2 membrane transloca-tion (*La Venuta et al., 2015*; *Steringer et al., 2012*, *2015*). The membrane inserted state of FGF2 oligomers has been proposed to be linked to a membrane pore with a toroidal architecture (*Steringer et al., 2012*). This view is supported by the observation that membrane lipids undergo transbilayer diffusion (*Steringer et al., 2012*). It appears possible that $PI(4,5)P_2$ molecules them-selves redistribute between monolayers when FGF2 oligomers insert into the membrane as it has been reported in other systems (*Bucki et al., 2000*). Membrane insertion of FGF2 oligomers is stimu-lated by phosphorylation of tyrosine 81 in FGF2 mediated by the non-receptor tyrosine kinase Tec, a *trans*-acting factor that is associated with the inner leaflet through an interaction of its PH domain with $PI(3,4,5)P_3$ (*Steringer et al., 2012*; *Ebert et al., 2010*; *La Venuta et al., 2016*). Recently, the integral membrane protein ATP1A1 has been identified as another *trans*-acting factor in unconven-tional secretion of FGF2 (*La Venuta et al., 2015*; *Zacherl et al., 2015*) FGF2 was shown to directly interact with the cytoplasmic domain of ATP1A1 (*La Venuta et al., 2015*; *Zacherl et al., 2015*), how-ever, the precise function of ATP1A1 in FGF2 secretion is unknown. In addition to the *cis* elements that are important for FGF2 binding to $PI(4,5)P_2$ (K127/R128/K133) and heparan sulfates (K133) as well as Y81 being the target of Tec kinase, membrane insertion of FGF2 oligomers depends on two cysteine residues (C77/C95) on the molecular surface of FGF2 (*La Venuta et al., 2015*; *Müller et al., 2015*). Intriguingly, despite a high degree of overall amino acid conservation among FGFs, C77 and C95 are absent from all members of the FGF family carrying signal peptides. C77/C95 have been shown to form intermolecular disulfide bridges, a process that drives efficient oligomerization of FGF2 in a $PI(4,5)P_2$-dependent manner (*Müller et al., 2015*). Following membrane insertion, cell sur-face heparan sulfate proteoglycans are required to complete membrane translocation by disassem-bling FGF2 oligomers at the outer leaflet and trapping of FGF2 in the extracellular space (*Nickel, 2007*; *Zehe et al., 2006*; *Seelenmeyer et al., 2005*). The interaction between FGF2 and cell surface heparan sulfates is mediated by basic residues in the C-terminal part of FGF2 with K133 being an essential component of this binding motif (*Nickel and Rabouille, 2009*, *2008*; *Temmerman et al., 2008*). Thus, four *cis*-elements in FGF2 [K127/R128/K133 forming the $PI(4,5)P_2$ binding pocket and K133 involved in FGF2 binding to both $PI(4,5)P_2$ and heparan sulfates, Y81 being the target of Tec kinase and C77/C95 promoting FGF2 oligomerization] and four *trans*-acting factors [$PI(4,5)P_2$, ATP1A1, Tec kinase and heparan sulfate proteoglycans] have been identi-fied to participate in unconventional secretion of FGF2 from cells. All known *trans*-acting factors are physically associated with the plasma membrane, the subcellular site of membrane translocation dur-ing unconventional secretion of FGF2 (*La Venuta et al., 2015*).

The molecular mechanism by which FGF2 is secreted from cells might be similar to other proteins secreted by unconventional means. This includes HIV-Tat that has been demonstrated to be secreted from infected T cells in a $PI(4,5)P_2$ dependent manner (*Debaisieux et al., 2012*; *Rayne et al., 2010a*, *2010b*). Consistently, HIV-Tat was recently found to bind to $PI(4,5)P_2$ concomitant with the formation of membrane pores (*Zeitler et al., 2015*). Interestingly, a recent study suggested a role

for membrane pores in unconventional secretion of Interleukin 1$\beta$ from macrophages (*Martín-Sánchez et al., 2016*). While Interleukin 1$\beta$ on its own cannot form pores in the plasma membrane, its release from macrophages might be coupled to another factor that becomes activated in an inflammasome-dependent manner, gasdermin (*Ding et al., 2016*). Since gasdermin is capable of forming membrane pores in a PI(4,5)P$_2$ dependent manner, it appears possible that at least one pathway of Interleukin 1$\beta$ secretion exists that is based upon membrane pores that are formed by gasdermin oligomers upon activation of inflammasomes. Therefore, the molecular mechanisms of unconventional secretion of FGF2, HIV-Tat and even Interleukin 1$\beta$ might be related in mechanistic terms.

In the current study, we tested key predictions of the FGF2 membrane translocation model as well as investigated the structure-function relationship of membrane-inserted FGF2 oligomers as intermediates in this process. Furthermore, we used atomistic molecular dynamics simulations to shed light on the initial steps of PI(4,5)P$_2$ triggered FGF2 oligomerization and membrane insertion. Using both biochemical and structural approaches, we demonstrate binding to FGF2 of PI(4,5)P$_2$ versus heparin to be mutually exclusive, a key aspect of the vectorial FGF2 membrane translocation model with heparan sulfates forming a molecular trap for FGF2 on cell surfaces. Furthermore, using giant unilamellar vesicles (GUVs) as a model system for the plasma membrane, we define the minimal molecular machinery required for FGF2 membrane translocation in a fully reconstituted inside-out system. These studies revealed only two *trans*-acting factors to be essential for FGF2 membrane translocation, PI(4,5)P$_2$ on GUV surfaces and long-chain heparins in the lumen of GUVs, the latter being used as mimetics of cell surface heparan sulfate proteoglycans. Consistently, the corresponding *cis*-elements required for FGF2 binding to PI(4,5)P$_2$ and heparan sulfates as well as the two cysteine residues required for oligomerization and membrane insertion of FGF2 were found essential for FGF2 membrane translocation. In addition, using various kinds of single molecule techniques, we demonstrate membrane-inserted forms of FGF2 to represent highly dynamic oligomers with a subunit number in the range of 8–12 FGF2 molecules. These studies were complemented by molecular dynamics simulations gaining the first insights into the initial steps of PI(4,5)P$_2$ induced oligomerization of FGF2. Most importantly, through simultaneous interactions of FGF2 monomers with multiple PI(4,5)P$_2$ molecules, a high affinity orientation of FGF2 was identified that favors FGF2 dimerization through a C95-C95 disulfide bridge along with additional electrostatic interactions within the dimerization interface. Our combined findings establish both the molecular basis and the minimal molecular machinery required for unconventional secretion of FGF2 from cells that consists of a surprisingly simple set of factors. Our findings further demonstrate the core mechanism of FGF2 membrane translocation to be thermodynamically driven by PI(4,5)P$_2$ dependent FGF2 oligomerization and membrane insertion without a requirement for additional energy sources such as ATP. The combination of PI(4,5)P$_2$ dependent FGF2 oligomerization, membrane insertion and heparan sulfate mediated trapping establishes a new type of self-sustained protein translocation across membranes that explains the molecular basis of how FGF2 is secreted from cells.

## Results

### Binding to FGF2 of PI(4,5)P$_2$ versus heparin is mutually exclusive

The current model describing the unconventional secretory pathway of FGF2 is based upon direct protein translocation across the plasma membrane. As a prerequisite for directional transport of FGF2 into the extracellular space, a key prediction of this model are sequential and mutually exclusive interactions of FGF2 with PI(4,5)P$_2$ at the inner leaflet and heparan sulfates at the outer leaflet. (*La Venuta et al., 2015*; *Nickel, 2011*; *Steringer et al., 2015*). Thus, employing both structural and biochemical methods, we tested whether interactions of FGF2 with PI(4,5)P$_2$ and heparin (mimicking cell surface heparan sulfates) are mutually exclusive. First, we conducted NMR spectroscopy to compare the binding epitopes in FGF2 for IP$_3$ [the headgroup of PI(4,5)P$_2$] and a defined heparin disaccharide suitable for NMR measurements (*Figure 1*). IP$_3$ and the heparin disaccharide were titrated with a variant form of FGF2 that is incapable of forming oligomers (FGF2-C77/95S [*Müller et al., 2015*]; 80 $\mu$M; $^{15}$N-isotope-labeled). As shown in *Figure 1A, B and C*, addition of 900 $\mu$M IP$_3$ to $^{15}$N-labeled FGF2-C77/95S (green contour lines) leads to large chemical shift changes for certain resonances as exemplified by G36, G135 and I145. Titration curves for individual residues within the

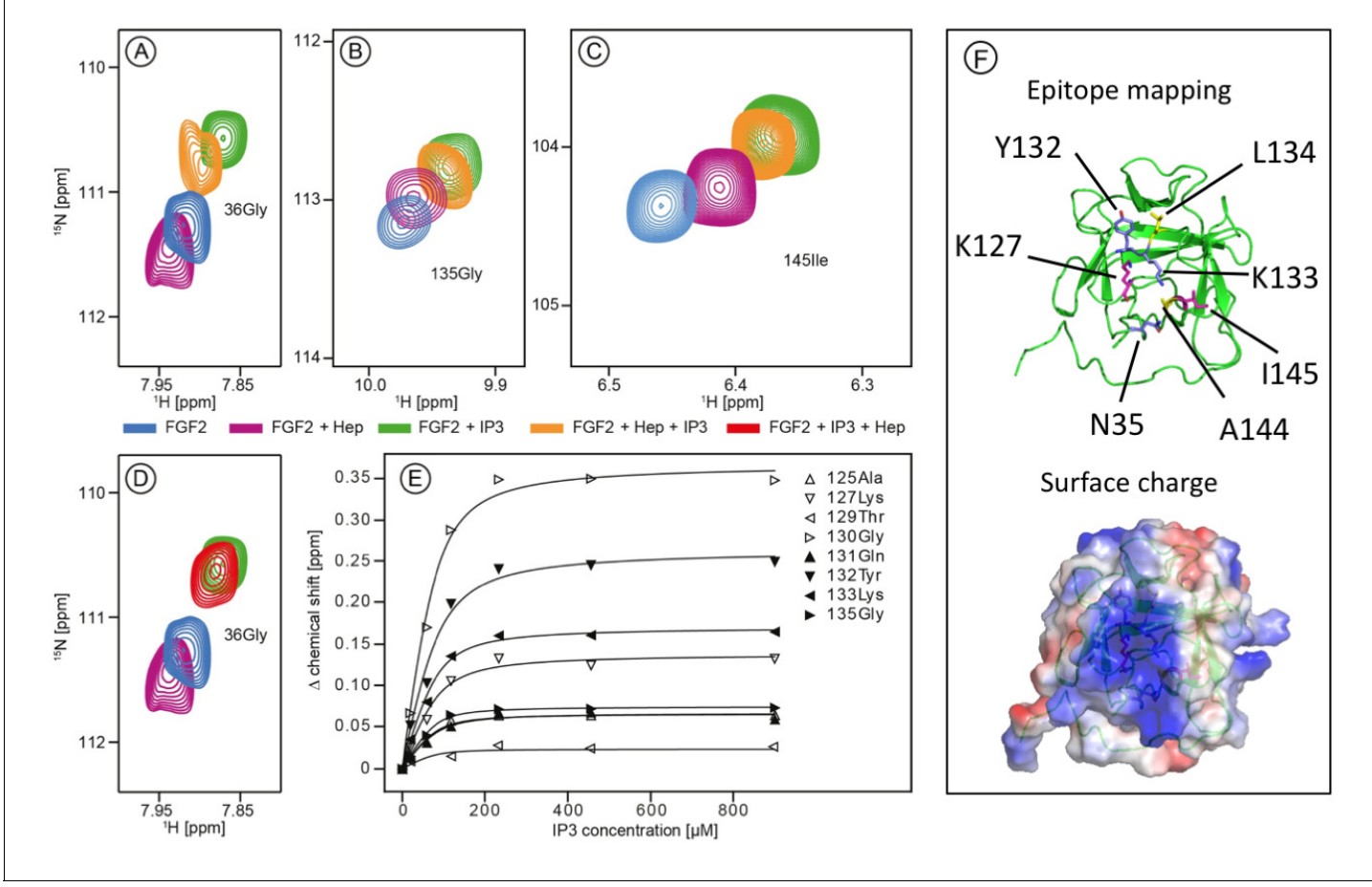

**Figure 1.** Structural analysis of the FGF2 binding epitopes for IP$_3$ [head group of PI(4,5)P$_2$] and heparin employing NMR spectroscopy. Enlarged regions of exemplary resonance peaks from a two-dimensional $^{15}$N-$^1$H correlation spectrum that are shifted upon addition of IP$_3$ and Heparin or both are shown on panels (A to –D). In panel (E), titration curves for individual $^{15}$N-$^1$H resonances are shown. Binding curves were fitted according to a simple two-state binding model and a K$_D$ was derived from the mean of the individual titration curves. HSQC for all NMR titration experiments are given in *Figure 1—source data 1*. In panel F (top), a cartoon of the FGF2 NMR structure is shown (PDB: 1BLD) with the side-chain of residues most significantly shifted upon IP$_3$ binding (blue), heparin binding (yellow) or affected by both binding partners (magenta). In addition, surface mapping of residues shown in panel E using the known structure (PDB: 1BLD) of FGF2 is illustrated in the bottom part of panel (F). Both IP$_3$ and heparin binding epitopes map to the same positively charged region of FGF2 highlighted in blue.

The following source data is available for figure 1:

**Source data 1.** Data for *Figure 1*, panels A-E.

binding epitope are depicted in *Figure 1E*. The mean K$_D$ value derived from eight resonances was determined as 16.2 ± 4.8 μM. This value is within a range similar to previous data using isothermal titration calorimetry with a K$_D$ value in the low micromolar range (*Temmerman et al., 2008*; *Temmerman and Nickel, 2009*). In comparison, addition of the small disaccharide variant of heparin led to relatively small changes of an overlapping set of resonances, indicating a weak interaction. For example, as shown in *Figure 1A, B and C* (purple contour lines), the resonances G36, G135 and I145 are affected by heparin, however, to a significantly smaller extent than the chemical shifts caused by IP$_3$ (green contour lines). Furthermore, titrating IP$_3$ to FGF2 with a pre-bound heparin disaccharide resulted in similar chemical shift changes as for the titration of IP$_3$ to FGF2 alone. This strongly indicates that IP$_3$ is outcompeting the heparin disaccharide under these conditions. Consistently, titrating the heparin disaccharide to FGF2 with a pre-bound IP$_3$ does not outcompete the IP$_3$ binding. *Figure 1D* shows that the signal of Gly36 of FGF2 is shifting in opposite directions upon IP$_3$

or heparin disaccharide binding, respectively. Addition of the heparin disaccharide to FGF2 with pre-bound $IP_3$ (red contour lines) does not shift the Gly36 signal from '$IP_3$' bound (green contour lines) to 'heparin bound' (purple contour lines). When replacing the heparin disaccharide by equimolar amounts of a mixture of long-chain heparins in NMR experiments we obtained a significant loss of signal intensity, indicating slow exchange and strong binding of long-chain heparins to FGF2 (data not shown). Based on the assignment of FGF2 (BMRB code 18995) the binding epitope could be identified as illustrated in *Figure 1F*. As anticipated, the two epitopes overlap and map to a region of the protein surface that is highly positively charged. These findings are in agreement with the known crystal structure of FGF2 bound to heparin (*Faham et al., 1996*).

The NMR data shown in *Figure 1* demonstrate a substantial overlap of the FGF2 binding epitopes for $PI(4,5)P_2$ and heparin and provide initial evidence for these interactions to be mutually exclusive. To challenge these findings using an independent method we conducted biochemical competition experiments. FGF2 was bound to liposomal membranes with a plasma-membrane-like lipid composition including $PI(4,5)P_2$ (*Figure 2*). Liposomes with FGF2 bound to $PI(4,5)P_2$ were treated with increasing concentrations of either long-chain heparin molecules (consisting of a mixture with various numbers of disaccharide units with high affinity towards FGF2 similar to heparan sulfates on cell surfaces; *Figure 2A and B*) or a defined low affinity heparin disaccharide (*Figure 2C and D*). For each condition, bound and unbound FGF2 was analyzed and quantified (*Figure 2B and D*). These experiments revealed that long-chain heparin molecules directly compete with $PI(4,5)P_2$ for binding to FGF2 with a half-maximal effect at a concentration of $\approx 5~\mu M$ (*Figure 2A and B*). By contrast, a defined heparin disaccharide was incapable of competing with $PI(4,5)P_2$ for binding to FGF2 (*Figure 2C and D*). These findings are consistent with the NMR experiments shown in *Figure 1D* where the low affinity heparin disaccharide fails to replace pre-bound $IP_3$ and provide direct proof for mutually exclusive interactions of FGF2 with $PI(4,5)P_2$ versus heparin. In a cellular context, similar to long-chain heparin molecules used in biochemical reconstitution experiments throughout this study, we conclude that high affinity heparan sulfates on cell surfaces [KD $\approx$ 100 nM; (*Faham et al., 1996*)] outcompete $PI(4,5)P_2$ [KD $\approx$ 5–15 $\mu M$; (*Temmerman et al., 2008*; *Temmerman and Nickel, 2009*) and *Figure 1B*] with regard to binding to FGF2. Thus, our data provide direct proof for a central prediction of the FGF2 membrane translocation hypothesis with sequential and competing interactions of FGF2 with $PI(4,5)P_2$ at the inner leaflet and cell surface heparan sulfates at the outer leaflet of the plasma membrane (*La Venuta et al., 2015*).

## Biochemical reconstitution of FGF2 membrane translocation with purified components

Based upon our findings demonstrating direct competition between $PI(4,5)P_2$ and long-chain heparins for binding to FGF2 (*Figures 1* and *2*), we aimed at reconstituting the minimal machinery of FGF2 membrane translocation with purified components (*Figure 3*). GUVs with a plasma-membrane-like lipid composition were generated that both expose $PI(4,5)P_2$ on their surfaces and contain long-chain heparins in their lumen. These GUVs also contained rhodamine-labelled phosphatidylethanolamine to allow for imaging the lipid bilayer. A phosphomimetic form of wild-type FGF2 [Y81pCMF; (*La Venuta et al., 2015*; *Steringer et al., 2012*; *Ebert et al., 2010*; *La Venuta et al., 2016*)] was used as a GFP fusion protein (FGF2-Y81pCMF-GFP) to monitor FGF2 membrane translocation into the lumen of GUVs. In addition, a small fluorescent tracer (Alexa647) was used to detect the formation of membrane pores (*La Venuta et al., 2015*; *Steringer et al., 2012*). In *Figure 3*, two experimental conditions were compared analyzing GUVs that either contained (*Figure 3A*) or lacked (*Figure 3B*) luminal long-chain heparin. In both cases, FGF2-Y81pCMF-GFP bound efficiently to the membrane surface due to the presence of $PI(4,5)P_2$. By contrast, only in case luminal heparin was included, FGF2-Y81pCMF-GFP membrane translocation into the lumen of GUVs was observed. This phenomenon was quantified based upon measuring GFP fluorescence intensity in the lumen, at the membrane and in the vesicle exterior as described in 'Methods'. In the given example, the ratio between luminal and external fluorescence was 4.93 in the presence of luminal heparin (Note increased fluorescence intensity on the luminal side in sub-panel e of panel A) and 1.33 in the absence of luminal heparin (Note equal fluorescence intensities on the luminal and external side in sub-panel e of panel B). In panels C and D of *Figure 3*, cross sections of 3D reconstructions are shown visualizing the dependence of FGF2-GFP membrane translocation on the presence of luminal heparin with a spatial view into the interior of GUVs. Based on a threshold of 1.6, a statistical analysis

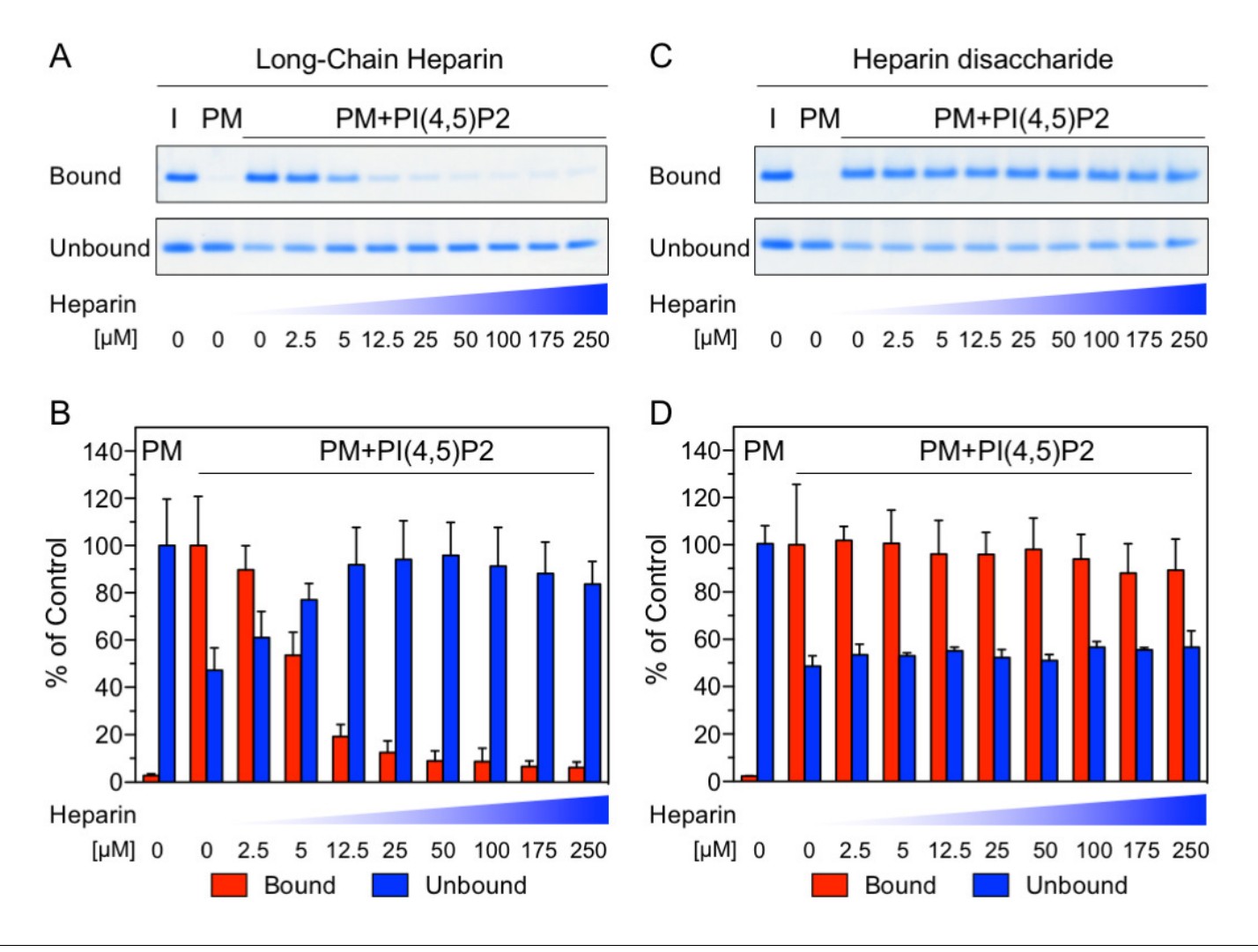

**Figure 2.** Binding of FGF2 to PI(4,5)P2 and heparin is mutually exclusive. Biochemical analysis employing plasma-membrane-like liposomes either lacking (PM) or containing 2 mol% PI(4,5)$P_2$ (PM +PIP2). Large unilamellar vesicles (LUVs) with bound His-FGF2-Y81pCMF-WT were incubated with increasing concentrations of either long-chain heparins or a defined heparin disaccharide as indicated. After 1 hr of incubation liposome-associated material (bound) was separated from supernatants (unbound). 50% of bound, 13.5% unbound and 14.8% of input material (I) were analyzed by SDS-PAGE. Coomassie-derived signals were quantified and normalized to controls. The fraction of FGF2-Y81pCMF-WT bound to PM-like liposomes containing PI(4,5)$P_2$ in the absence of heparin was set to 100% (red bars in **Figure 2B and D**). The unbound fraction of FGF2-Y81pCMF-WT was normalized using PM-like liposomes lacking PI(4,5)$P_2$ (blue bars in **Figure 2B and D**). Mean values with standard deviations are shown (n = 3). Raw and normalized data of individual experiments as well as calculations of mean values with standard deviations are shown in **Figure 2—source data 1**.

The following source data is available for figure 2:

**Source data 1.** Data for **Figure 2**, panels B and D.

of 20 to 120 GUVs from at least three independent experiments was conducted providing the percentages of GUVs with FGF2-Y81pCMF-GFP being enriched in their lumen for each experimental condition. This analysis revealed that FGF2-Y81pCMF-GFP membrane translocation occurred in about 23% of the GUVs with luminal heparin (Figure 9). By contrast, less than 2% of GUVs without luminal heparin contained FGF2-Y81pCMF-GFP in their lumen (Figure 9). Simultaneously, the small fluorescent tracer Alexa647 was used to monitor membrane pore formation as indicated by equal fluorescence intensities in the lumen and the exterior of GUVs. Under the two conditions shown in **Figure 3**, membrane pore formation did not depend on the presence of heparin in the lumen of

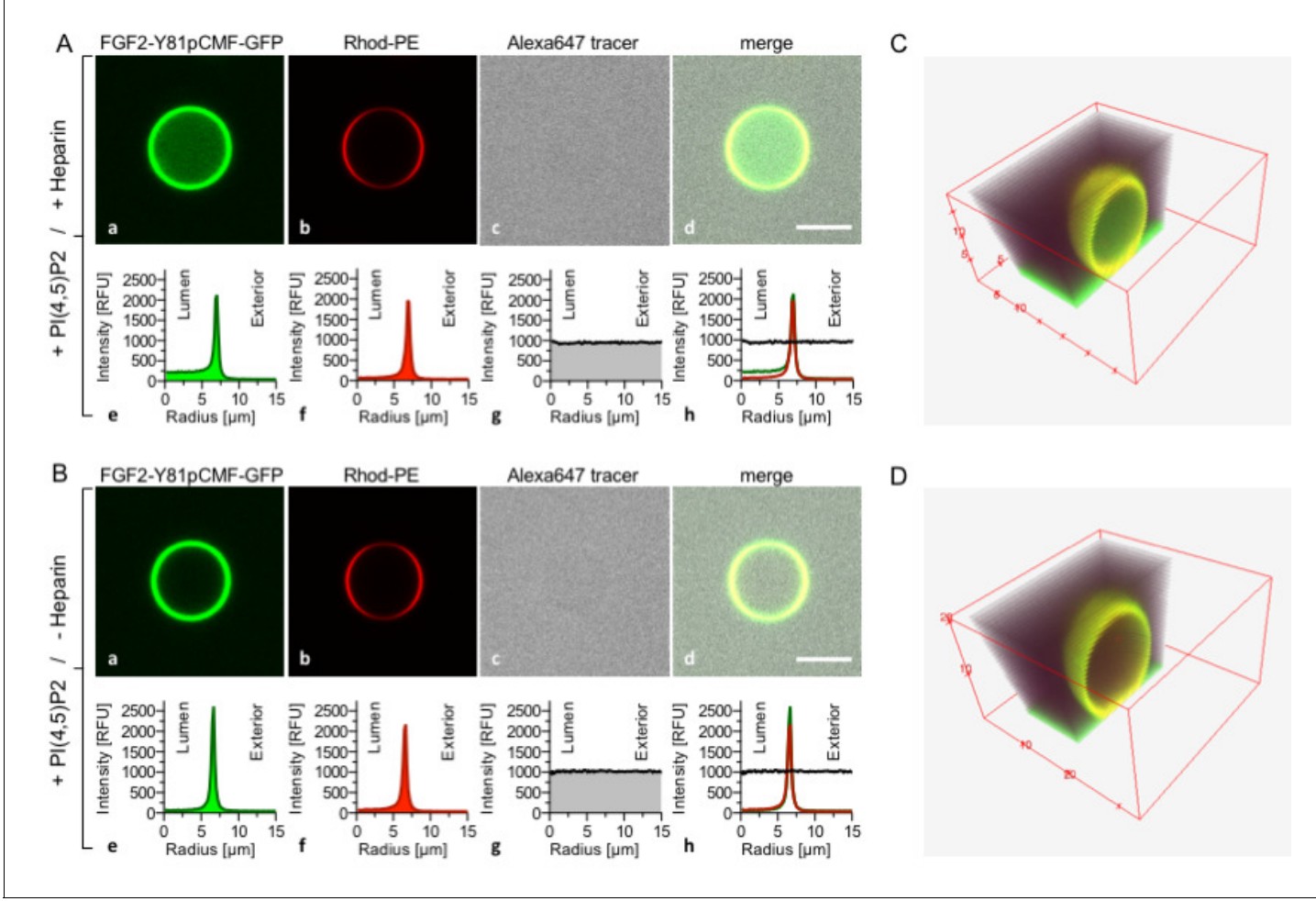

**Figure 3.** Reconstitution of FGF2 membrane translocation with purified components. Giant unilamellar vesicles with a plasma membrane-like lipid composition containing PI(4,5)P$_2$ were prepared in the presence (panel **A** and **C**) or absence (panel **B** and **D**) of long-chain heparins. Rhodamine-PE was incorporated into the lipid bilayer during GUV preparation as membrane marker. After removal of excess heparin by low speed centrifugation, GUVs were incubated with FGF2-Y81pCMF-GFP (200 nM) and a small fluorescent tracer (Alexa647). Following 180 min of incubation luminal penetration of GUVs by FGF2-Y81pCMF-GFP and small tracer molecules was analyzed by confocal microscopy (scale bar = 10 µm). GUVs were analyzed in all three channels using the plugin 'Radial profile' of the ImageJ software as explained under 'Materials and methods'. Profile plots of normalized integrated intensities around concentric circles as a function of distance from the center of the GUV are given in relative fluorescence units (RFU). FGF2 membrane translocation is indicated by increased GFP fluorescence intensity in the lumen of GUVs compared to the exterior as exemplified in sub-panel e of panel **A**. The dependence of FGF2-Y81pCMF-GFP membrane translocation on luminal heparin is further documented by 3D reconstruction images (panels **C** and **D**) providing a spatial view into the GUV interior.

GUVs. As shown in Figure 9, membrane pore formation occurred in about 33% to 41% of all GUVs, both in the absence and the presence of luminal heparin. These results are consistent with previous findings demonstrating that membrane insertion of FGF2 oligomers concomitant with pore formation does not depend on luminal heparin (*Steringer et al., 2012*). GUVs containing luminal FGF2-Y81pCMF-GFP in the absence of membrane pore formation were undetectable.

To challenge the results shown in *Figure 3*, we conducted experiments where high-affinity long-chain heparins were directly compared with the low-affinity heparin disaccharide introduced in *Figures 1* and *2*. While FGF2-Y81pCMF-GFP membrane binding, membrane pore formation (Alexa647 intensity ratio lumen/exterior = 0.96) and FGF2 membrane translocation (GFP intensity ratio lumen/exterior = 6.44) was fully functional for GUVs containing luminal long-chain heparins (*Figure 4A*), FGF2-Y81pCMF-GFP membrane translocation was impaired for GUVs containing the heparin disaccharide (*Figure 4B*; GFP intensity ratio lumen/exterior = 1.07). However, as expected, membrane

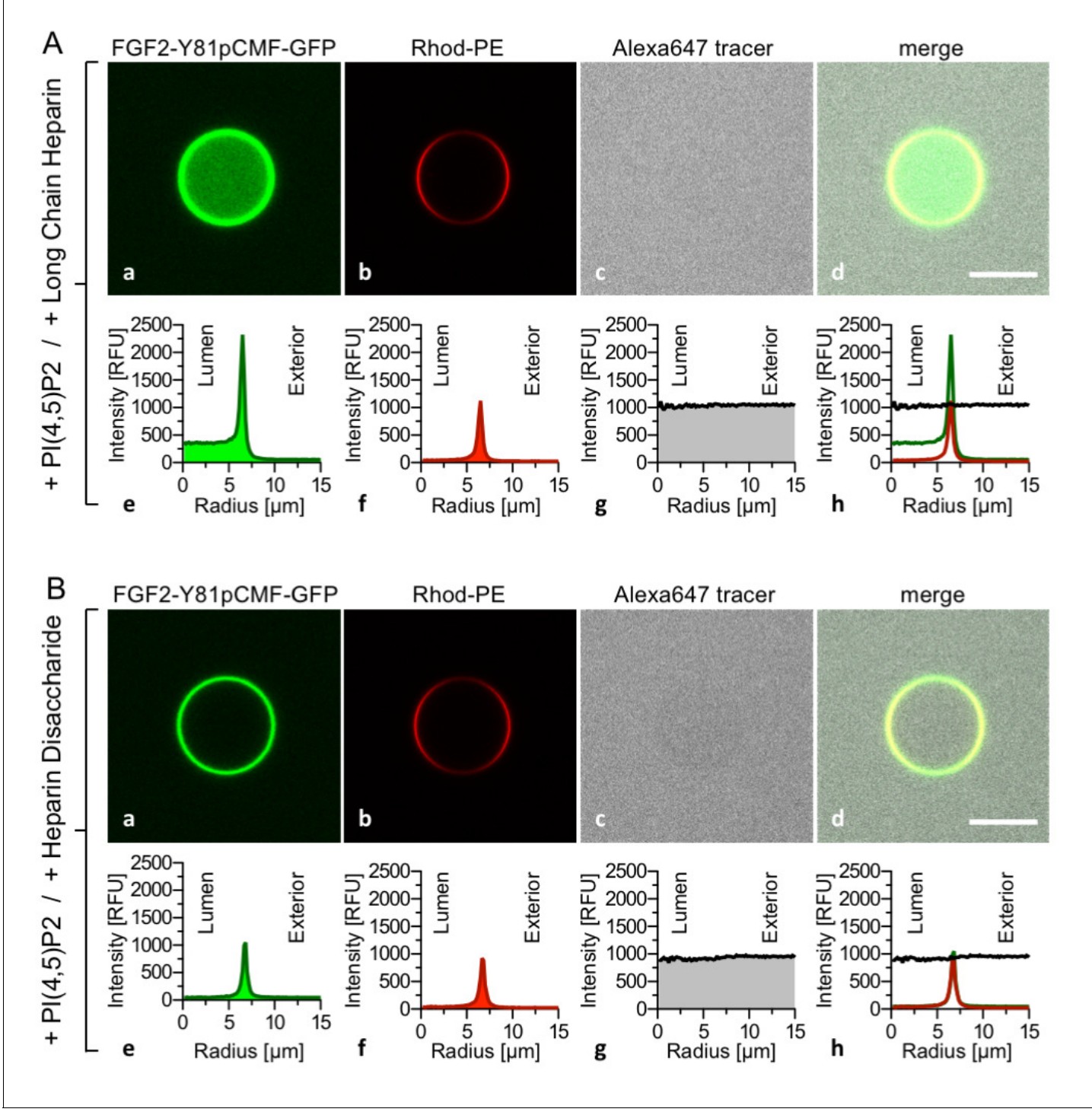

**Figure 4.** FGF2 membrane translocation depends on luminal long-chain heparins that cannot be substituted by low affinity heparin disaccharides. Giant unilamellar vesicles with a plasma membrane-like lipid composition containing PI(4,5)P₂ were prepared that contain either long-chain heparins (panel **A**) or a defined heparin disaccharide (panel **B**). Incubation with FGF2-Y81pCMF-GFP and data analysis were conducted as described in the legend to *Figure 3* and under 'Materials and methods'.

binding and pore formation was normal under these conditions (*Figure 4B*; Alexa647 intensity ratio lumen/exterior = 0.95). As depicted in Figure 9, a statistical analysis revealed that substitution of luminal long-chain heparins by a heparin disaccharide causes a drop from about 23% to only 5% of GUVs positive for FGF2-Y81pCMF-GFP membrane translocation. Our combined findings from *Figures 3* and *4* establish FGF2 membrane translocation in a minimal system employing GUVs with a plasma-membrane-like lipid composition including $PI(4,5)P_2$ and luminal long-chain heparins mimicking heparan sulfates from cell surfaces.

## Dependence of FGF2 membrane translocation on $PI(4,5)P_2$

To analyze a requirement for $PI(4,5)P_2$ on FGF2-Y81pCMF-GFP induced membrane pore formation and membrane translocation (*Figure 5*), four types of GUVs were used all of which contained luminal long-chain heparins but differed with regard to lipid composition and the presence of $PI(4,5)P_2$ on their membrane surfaces. Similar to the experiments shown in *Figures 3* and *4*, FGF2-Y81pCMF-GFP membrane binding, membrane pore formation (Alexa647 intensity ratio lumen/exterior = 0.97) and FGF2 membrane translocation (GFP intensity ratio lumen/exterior = 3.27) could be observed when GUVs contained $PI(4,5)P_2$ on their membrane surface (*Figure 5A*). By contrast, when cholesterol was omitted from the plasma membrane like lipid composition, $PI(4,5)P_2$ dependent membrane recruitment was significantly impaired (*Figure 5B*). This observation is consistent with earlier findings suggesting an impact of cholesterol dependent microdomain formation on the efficiency of FGF2 binding to $PI(4,5)P_2$ (17,33). In turn, under these conditions, FGF2-Y81pCMF-GFP membrane pore formation (*Figure 5B*; Alexa647 intensity ratio lumen/exterior = 0.05) and FGF2 membrane translocation (*Figure 5B*; GFP intensity ratio lumen/exterior = 0.84) could not be observed. These findings were corroborated by a statistical analysis depicted in Figure 9 revealing that only about 6% of GUVs were positive for FGF2-Y81pCMF-GFP membrane translocation when cholesterol was removed from a plasma membrane like lipid composition that included $PI(4,5)P_2$.

In the absence of $PI(4,5)P_2$ on the surface of GUVs with luminal long-chain heparins, a complete failure of FGF2-Y81pCMF-GFP membrane recruitment was observed (*Figure 5C*). This in turn resulted in a lack of membrane pore formation (Alexa647 intensity ratio lumen/exterior = 0.05) and FGF2-Y81pCMF-GFP translocation into the lumen of GUVs (GFP intensity ratio lumen/exterior = 0.7). As shown in Figure 9, in the absence of $PI(4,5)P_2$, less than 2% of GUVs were characterized by membrane pores and FGF2 membrane translocation was undetectable (Figure 9). The fourth type of GUVs in this set of experiments was characterized by substitution of $PI(4,5)P_2$ with a Ni-NTA lipid (*Figure 5D*). This experimental condition has previously been demonstrated to allow for efficient recruitment of His-tagged FGF2 to membrane surfaces. However, under these conditions, a substantial reduction of membrane pore formation was observed (*Steringer et al., 2012*). This phenotype is not due to a general block in FGF2 oligomerization but is likely to be related to a requirement for $PI(4,5)P_2$ to facilitate FGF2-dependent formation of a toroidal membrane pore that is characterized by strong membrane curvature (*La Venuta et al., 2015*). Indeed, in the example given in *Figure 5D*, the tracer intensity ratio (lumen/exterior) was found to be 0.05 demonstrating pore formation to depend on $PI(4,5)P_2$, even under conditions where membrane recruitment of FGF2-Y81pCMF-GFP is mediated by other means. Consistently, in the statistical analysis shown in Figure 9, only about 10% of GUVs containing the Ni-NTA lipid to recruit His-tagged FGF2-GFP were found to contain membrane pores. This phenotype was accompanied by an almost complete failure of His-tagged FGF2-Y81pCMF-GFP to translocate into the lumen of GUVs when $PI(4,5)P_2$ was substituted by a Ni-NTA lipid (GFP intensity ratio lumen/exterior = 1.15 in *Figure 5D*). As shown in Figure 9, this was confirmed by a statistical analysis with less than 1% of Ni-NTA GUVs containing luminal FGF2-Y81pCMF-GFP.

Our combined findings from *Figures 3*, *4* and *5* along with their quantification given in Figure 9 define the minimal molecular machinery required for FGF2 membrane translocation. Two *trans*-acting factors that previously have been demonstrated to be required for FGF2 secretion from cells (*La Venuta et al., 2015*; *Temmerman et al., 2008*; *Zehe et al., 2006*) are essential for this process, $PI(4,5)P_2$ and long-chain heparins, the latter mimicking cell surface heparan sulfates.

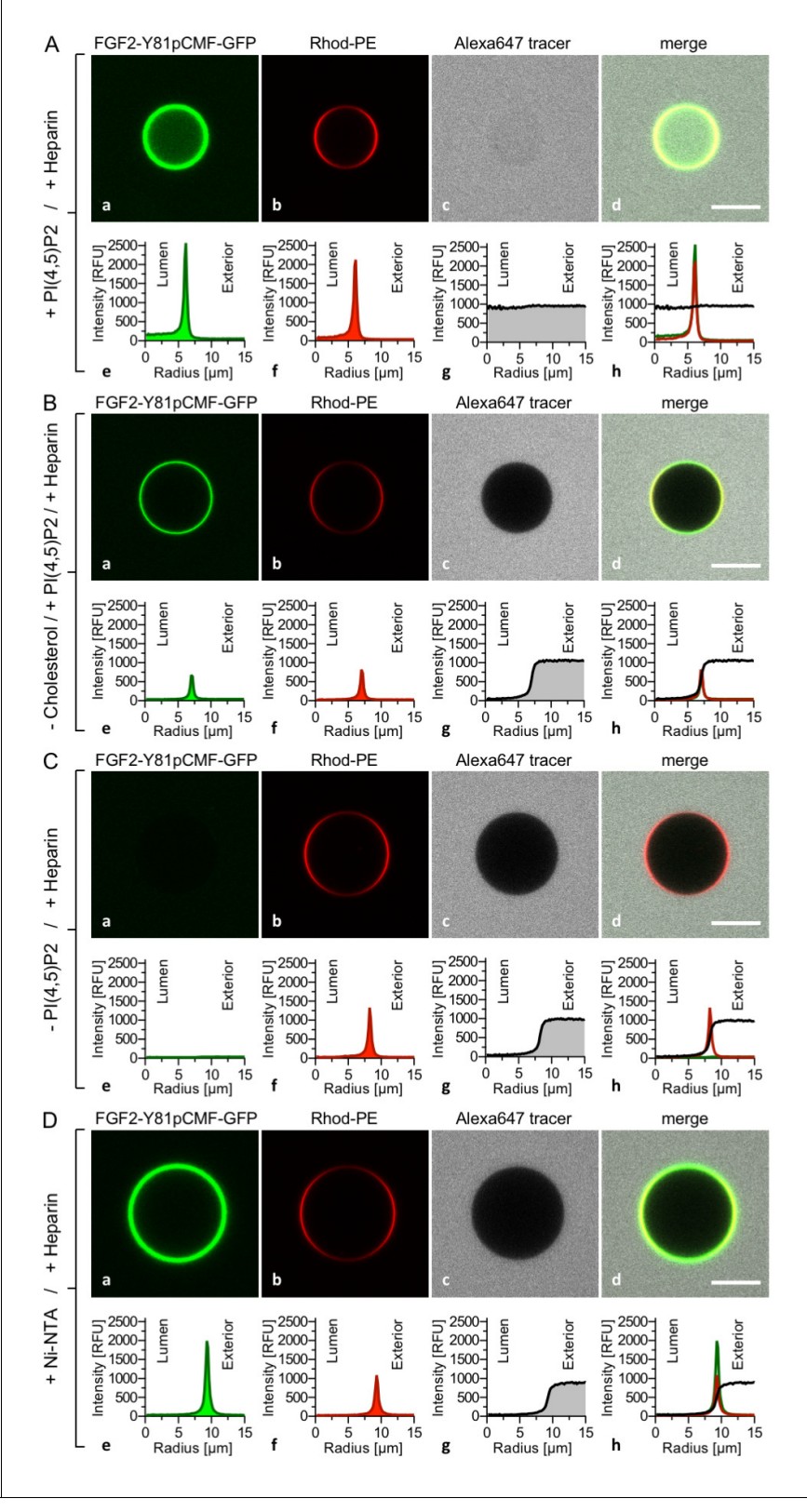

**Figure 5.** PI(4,5)P$_2$ is required for both membrane pore formation and FGF2 membrane translocation. Giant unilamellar vesicles were prepared with a plasma membrane-like lipid composition either containing PI(4,5)P$_2$ (panel **A**), containing PI(4,5)P$_2$ but lacking cholesterol (panel **B**), lacking PI(4,5)P$_2$ (panel **C**) or containing a Ni-NTA lipid substituting PI(4,5)P$_2$ (panel **D**). All four types of GUVs contained luminal long-chain heparins. Incubation with

*Figure 5 continued on next page*

*Figure 5 continued*

FGF2-Y81pCMF-GFP and data analysis were conducted as described in the legend to *Figure 3* and under 'Materials and methods'. Note increased GFP fluorescence in the lumen of GUVs as exemplified in sub-panel e of panel A indicating FGF2-Y81pCMF-GFP membrane translocation.

## Dependence of FGF2 membrane translocation on cis-elements in FGF2

Following the analysis of *trans*-acting factors for FGF2 membrane translocation in a minimal system, we aimed at studying *cis*-elements in FGF2 known to be required for FGF2 secretion from cells (*La Venuta et al., 2015*). We used three variant forms of purified FGF2-Y81pCMF-GFP that either have a defect in binding to $PI(4,5)P_2$ [K127Q/R128Q; (*Nickel, 2011*; *Temmerman et al., 2008*; *Temmerman and Nickel, 2009*)], a defect in binding to both $PI(4,5)P_2$ and long-chain heparins [K127Q/R128Q/K133Q; (*Nickel, 2011*; *Temmerman et al., 2008*; *Temmerman and Nickel, 2009*)] or a defect in oligomerization and membrane pore formation [C77A/C95A; (*La Venuta et al., 2015*; *Müller et al., 2015*)]. The various forms of FGF2-Y81pCMF-GFP were characterized with regard to binding to both long-chain heparins immobilized on beads (*Figure 6A and B*) and $PI(4,5)P_2$ as part of liposomes using a flow cytometry setup [*Figure 6C*; (*Temmerman et al., 2008*; *Temmerman and Nickel, 2009*)]. In addition, the processed form of interleukin 1$\beta$ fused to GFP (IL-1$\beta$-GFP) and GFP alone were taken along as control proteins. Binding of the various proteins to long-chain heparins and $PI(4,5)P_2$ was compared to the wild-type form of FGF2-Y81pCMF-GFP. As shown in *Figure 6A and B*, substitution of C77 and C95 by alanines (FGF2-Y81pCMF-C77/95A-GFP) did not affect binding to long-chain heparins. By comparison, FGF2-Y81-pCMF-C77/95A-GFP showed reduced binding to $PI(4,5)P_2$ (*Figure 6C*). This is caused by a reduction in binding avidity due to the inability of this FGF2 variant form to oligomerize. By contrast, the K127Q/R128Q/K133Q form of FGF2-Y81pCMF-GFP did neither bind to long-chain heparins (*Figure 6A and B*) nor to $PI(4,5)P_2$ (*Figure 6C*). A differential phenotype was observed for the K127Q/R128Q of FGF2-Y81pCMF-GFP which did bind to long-chain heparins (*Figure 6A and B*) but failed to interact with $PI(4,5)P_2$ (*Figure 6C*). Finally, neither IL-1$\beta$-GFP nor GFP alone did bind to long-chain heparins or $PI(4,5)P_2$ (*Figure 6A, B and C*).

The FGF2 variant forms and control proteins described above were tested for activity in membrane pore formation and membrane translocation using GUVs containing both $PI(4,5)P_2$ and long-chain heparins in their lumen (*Figures 7* and *8*). The wild-type form of FGF2-Y81pCMF-GFP was used as a reference in both assays (*Figures 7A* and *8A*; GFP intensity ratio lumen/exterior = 7.14 and 6.3, respectively). As expected, neither the K127Q/R128Q/K133Q form nor the K127Q/R128Q form of FGF2-Y81pCMF-GFP were capable of binding to $PI(4,5)P_2$-containing GUVs. Consistently, neither the K127Q/R128Q/K133Q (GFP intensity ratio lumen/exterior = 0.61; *Figure 7B*) nor the K127Q/R128Q form (GFP intensity ratio lumen/exterior = 0.6; *Figure 7C*) of FGF2-Y81pCMF-GFP were capable of translocating into the lumen of GUVs containing both $PI(4,5)P_2$ and long-chain heparins. This was confirmed by the statistical analysis shown in *Figure 9* with membrane translocation being undetectable for K127Q/R128Q/K133Q and below 2% for K127Q/R128Q, respectively. As shown in the example given in *Figure 7D*, the C77/95A form of FGF2-Y81pCMF-GFP was characterized by a defect in membrane pore formation (Alexa647 intensity ratio lumen/exterior = 0.05) using GUVs containing both $PI(4,5)P_2$ and long-chain heparins. Similarly, FGF2-Y81pCMF-C77/95A-GFP translocation into the lumen was not observed (GFP intensity ratio lumen/exterior = 0.97; *Figure 7D*). As shown in *Figure 9*, about 10% of GUVs incubated with FGF2-Y81pCMF-C77/95A-GFP contained membrane pores and luminal FGF2-Y81pCMF-C77/95A-GFP, a significant reduction compared to FGF2-Y81pCMF-GFP. Finally, we tested two control proteins, IL-1$\beta$-GFP (*Figure 8B*) and GFP alone (*Figure 8C*). Both of them did not show any activity with regard to binding to $PI(4,5)P_2$, membrane pore formation (Alexa647 intensity ratio lumen/exterior = 0.03 and 0.05, respectively) and membrane translocation into the lumen of GUVs (GFP intensity ratio lumen/exterior = 0.56 and 0.48, respectively). These findings were confirmed by the statistical analysis shown in *Figure 9*. In conclusion, in addition to the *trans*-acting factors $PI(4,5)P_2$ and long-chain heparins as mimetics of cell surface heparan sulfates, FGF2 membrane translocation depends on *cis*-elements that allow for binding to $PI(4,5)P_2$ (K127/R128), binding to long-chain heparins (K133) and are required for oligomerization and membrane pore formation (C77/C95).

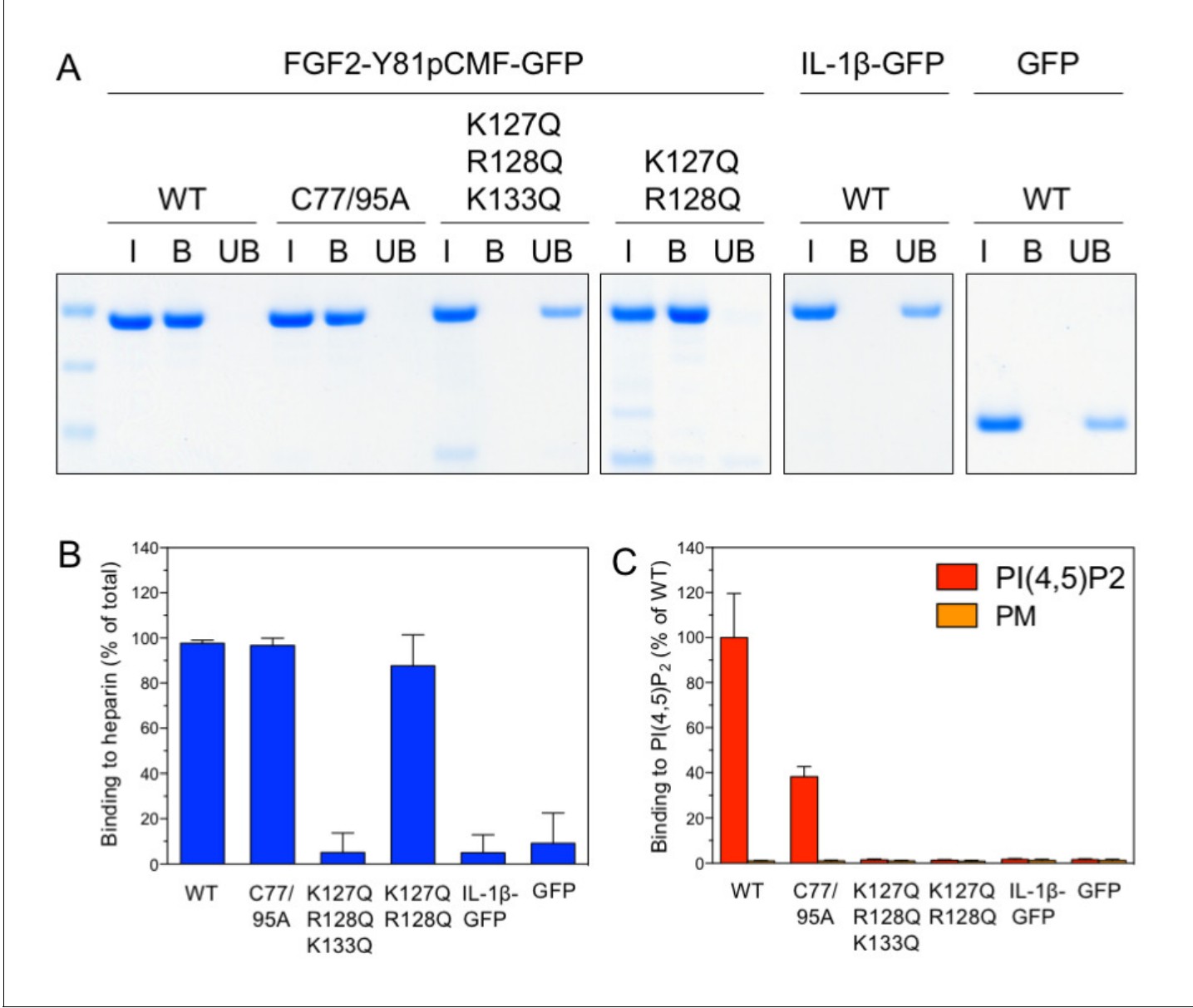

**Figure 6.** Analysis of FGF2 variant forms with differential defects in binding to PI(4,5)P$_2$ and heparin. The variant forms of FGF2-Y81pCMF-GFP indicated were tested for binding to heparin beads (panels **A** and **B**) and PI(4,5)P$_2$ (panel **C**). Heparin sepharose beads were incubated with the FGF2 fusion proteins indicated. Bound and unbound material was separated by centrifugation. Bound proteins were eluted with SDS sample buffer and analysed by SDS-PAGE and Coomassie staining [5% input (I), 20% bound (B) and 5% unbound (UB)]. Signals were quantified using a Li-COR Odyssey infrared imaging system. Mean values with standard deviations of three independent experiments are shown (panel **B**). Raw and normalized data of individual experiments as well as calculations of mean values with standard deviations are shown in *Figure 6—source data 1*. Binding of the FGF2 fusion proteins to PI(4,5)P$_2$ contained in plasma membrane-like liposomes was assessed using a flow-cytometry assay (*Temmerman et al., 2008*; *Temmerman and Nickel, 2009*) (panel **C**). Data were normalized by defining binding of FGF2-Y81pCMF-GFP to PI(4,5)P$_2$ as 100% binding efficiency. Mean values with standard deviations are shown (n = 4). Consider *Figure 6—source data 1* for more details.

The following source data is available for figure 6:

**Source data 1.** Data for *Figure 6*, panels B and C.

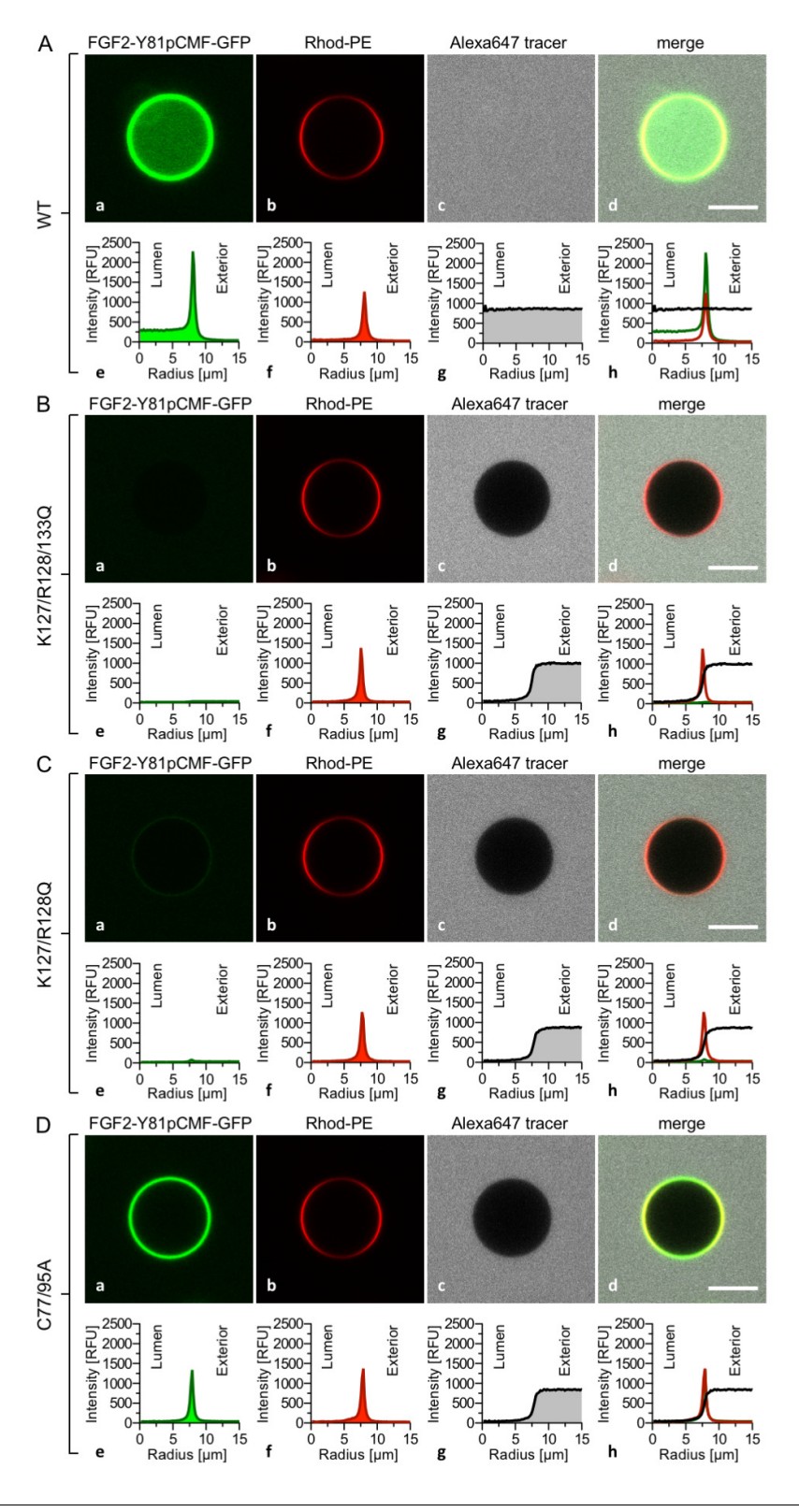

**Figure 7.** FGF2 membrane translocation depends on *cis*-elements mediating binding to PI(4,5)P$_2$ and heparin as well as driving FGF2 oligomerization and membrane pore formation. Giant unilamellar vesicles with a plasma membrane-like lipid composition containing both PI(4,5)P$_2$ and luminal long-chain heparins were prepared as described in the legend to *Figure 3* and under 'Materials and methods'. GUVs were incubated with variant forms of FGF2-Y81pCMF-GFP as indicated. These included the wild-type form (panel **A**), the K127Q/R128Q/K133Q form deficient in binding to PI(4,5)P$_2$ and

*Figure 7 continued on next page*

Figure 7 continued

heparin (panel **B**), the K127Q/R128Q form deficient in binding to PI(4,5)P$_2$ (panel **C**) and the C77A/C95A form deficient in oligomerization and membrane pore formation (panel **D**). Incubation conditions and data analysis were conducted as described in the legend to *Figure 3* and under 'Materials and methods'. Note increased GFP fluorescence in the lumen of GUVs as exemplified in sub-panel e of panel A indicating membrane translocation of the wild-type form of FGF2-Y81pCMF-GFP.

## Oligomeric state of membrane inserted FGF2 translocation intermediates in supported lipid bilayers

Beyond the reconstitution of FGF2 membrane translocation with purified components, we aimed at insight into the structure function relationship of membrane-inserted FGF2 oligomers, the key intermediates in unconventional secretion of FGF2 from cells. In particular, we analyzed the subunit number of membrane associated FGF2 oligomers. In a first approach, we used supported lipid bilayers (SLB) containing PI(4,5)P$_2$ to determine the oligomeric state of membrane inserted FGF2 complexes. To obtain high contrast single molecule detection we used FGF2 variant forms carrying a Halo-tag. This allowed for protein labeling with a bright and photo-stable fluorophore (Abberior StarRed) with a degree of labeling of 0.9 per FGF2 monomer. A three-step protocol was used to analyze the oligomeric state distribution of FGF2 translocation clusters (*Figure 10*; for details see 'Materials and methods'). Two variant forms of FGF2, FGF2-Y81pCMF-HALO-StarRed and FGF2-Y81pCMF-C77/95A-HALO-StarRed were loaded onto SLBs containing 2 mol% PI(4,5)P$_2$ at a final concentration of 100 nM. While FGF2-Y81pCMF-HALO-StarRed efficiently bound to SLBs, (*Figure 10D*), binding of FGF2-Y81pCMF-C77/95A-HALO-StarRed was significantly reduced (*Figure 10J*). These findings are consistent with the data shown in *Figures 6* and *7* and reflect the inability of FGF2-Y81pCMF-C77/95A-HALO-StarRed to oligomerize (see above). In a second step, following 10 min of incubation, SLBs were washed with a buffer containing 150 mM NaCl to remove unbound proteins (*Figure 10B,E,H and K*). At this stage, fluorescence recovery after photo-bleaching (FRAP) was used to determine the mobility of membrane bound FGF2 species (FRAP data available in *Figure 11—source data 1*). This analysis revealed the majority (98%) of FGF2-Y81pCMF-HALO-StarRed to be highly mobile with a diffusion constant of D = 0.4 μm$^2$/s. These findings indicate that after a short incubation with SLBs of only 10 min, most of FGF2-Y81pCMF-HALO-StarRed did not oligomerize into higher order structures which were integrated in the membrane. In a third step, the highly mobile fraction of FGF2-Y81pCMF-HALO-StarRed monomers bound to PI(4,5)P$_2$ was removed by high salt treatment (500 mM NaCl) (*Figure 10C,F,I and L*). The remaining population of the protein was largely immobile suggesting membrane insertion of oligomers with contacts to the glass support of SLBs. In contrast, we did not observe significant immobilization of FGF2-Y81pCMF-C77/95A-HALO-StarRed which is due to a failure of oligomerization and membrane insertion (*Müller et al., 2015*).

In order to determine the oligomeric state of the immobilized FGF2-Y81pCMF-HALO-StarRed clusters, 2D confocal images were recorded with a long pixel dwell-time of 1 ms. This resulted in high signal to background ratio images without significant photo-bleaching (*Figure 11A*). Single molecule localization was used to automatically determine the brightness of each cluster via 2D Gaussian fitting. The brightness of a single HALO-StarRed was determined by following the same imaging and fitting procedure with immobilized mEGFP-HALO-StarRed (data not shown). Finally, the number of monomers in each immobilized FGF2 cluster was estimated by normalizing the cluster brightness to the brightness of a single HALO-StarRed (*Figure 11A*). A Gaussian mixture analysis in MATLAB was used to estimate the number of sub-populations in the whole distribution of more than 1000 clusters from six independent samples. The fit yielded four components with 3, 6, 11 and 17 monomers per cluster, respectively, with the majority of the membrane inserted population after 10 min of incubation being in the trimeric and hexameric state (*Figure 11B*).

## Functional correlation of the oligomeric state of membrane associated FGF2 translocation intermediates and membrane pore formation

Beyond our studies using SLBs (see above) we aimed at using an experimental system that provides direct insight into the structure-function relationship of membrane inserted FGF2 translocation

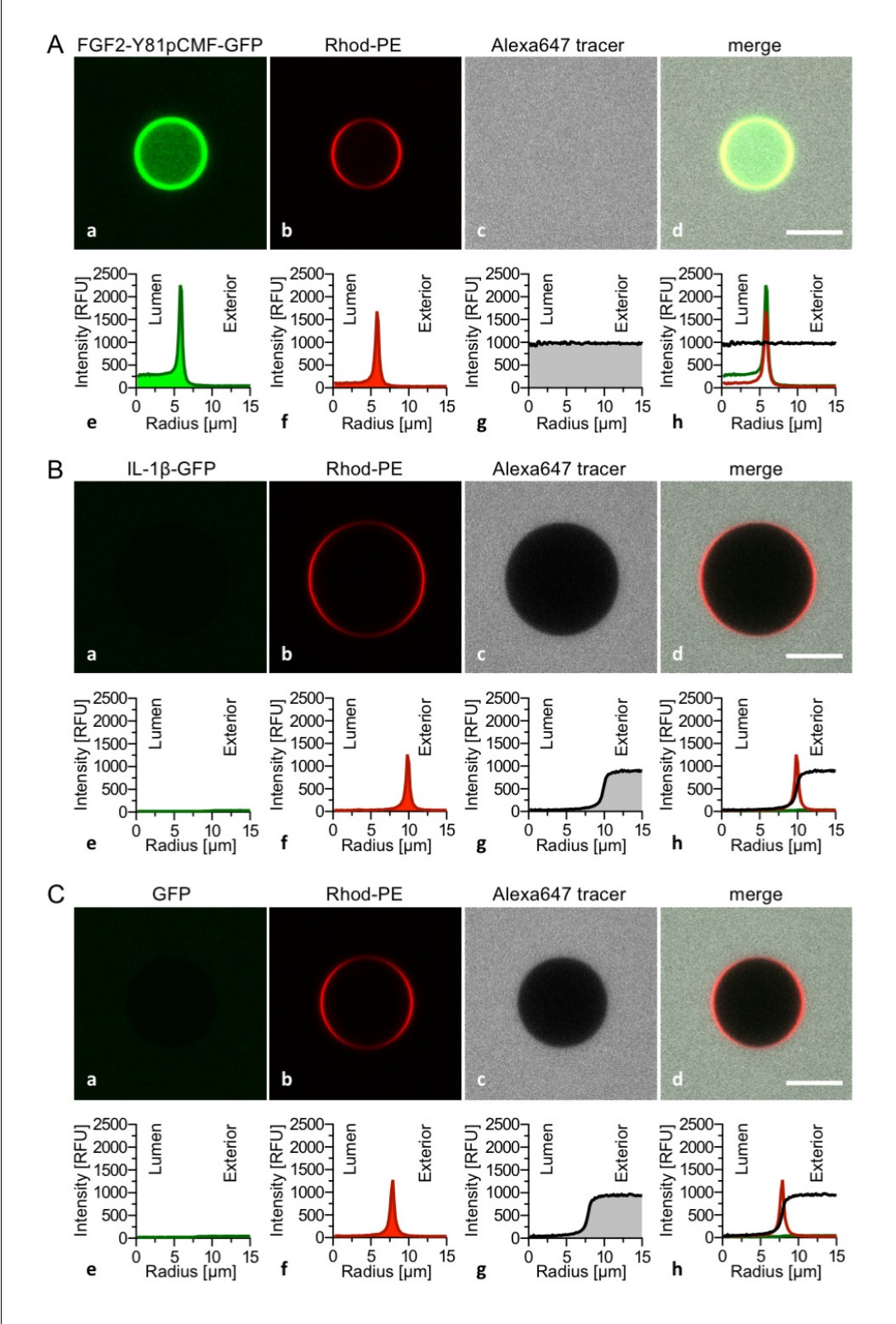

**Figure 8.** Interleukin 1$\beta$, a structural homologue of FGF2, is incapable of binding to PI(4,5)P$_2$, membrane pore formation and membrane translocation. Giant unilamellar vesicles with a plasma membrane-like lipid composition containing both PI(4,5)P$_2$ and luminal long-chain heparins were prepared as described in the legend to *Figure 3* and under 'Materials and methods'. Incubation conditions using the wild-type form of FGF2-Y81pCMF-GFP (panel A), the mature form of Interleukin 1$\beta$-GFP (panel B) and GFP as control protein (panel C) as well as data analysis were conducted as described in the
*Figure 8 continued on next page*

*Figure 8 continued*

legend to *Figure 3* and under 'Materials and methods'. Note increased GFP fluorescence in the lumen of GUVs as exemplified in sub-panel e of panel A indicating membrane translocation of the wild-type form of FGF2-Y81pCMF-GFP.

intermediates. Therefore, we conducted experiments to directly correlate oligomerization of FGF2-Y81pCMF-GFP with membrane pore formation in GUVs with a plasma membrane like lipid composition. Using z-scan fluorescence correlation spectroscopy [FCS; (*Benda et al., 2003*)], we combined a brightness and diffusion analysis of individual FGF2 oligomers with the membrane pore assay introduced in *Figure 3* (*Steringer et al., 2012*). The average oligomeric state (*Figure 12A, C and E*) and diffusion constants (*Figure 12B, D and F*) of membrane associated FGF2-Y81pCMF-GFP of individual GUVs were plotted as a function of protein concentration on membrane surfaces [$c$(FGF2-Y81pCMF-GFP)]. In addition, each GUV was classified regarding membrane pore formation based upon luminal penetration of a small fluorescent tracer. GUVs without membrane pores were characterized by a low average surface protein concentration of $c$(FGF2-Y81pCMF-GFP)=0.17 ± 0.27 nmol/m$^2$ (*Figure 12A*, open circles). By contrast, GUVs containing membrane pores were found to have a four-fold higher average protein surface concentration of $c$(FGF2-Y81pCMF-GFP)=0.71 ± 0.61 nmol/m$^2$ (*Figure 12C*; closed circles). The average oligomeric state measured on GUVs containing membrane pores was 9.23 ± 2.96 for $c$(FGF2-Y81pCMF-GFP) larger than 0.4 nmol/m$^2$ (57% of GUVs) and 5.24 ± 3.5 for $c$(FGF2-Y81pCMF-GFP) smaller than 0.4 nmol/m$^2$ (43% of GUVs). By contrast, 90% of the GUVs without membrane pores were characterized by an average oligomeric state of 5.22 ± 2.86 with $c$(FGF2-Y81pCMF-GFP)≤0.4 nmol/m$^2$. These experiments revealed a clear correlation between $c$(FGF2-Y81pCMF-GFP), the formation of higher oligomers and the probability of membrane pore formation.

In order to verify the determination of average oligomeric state values derived from brightness analyses, diffusion measurements were conducted for FGF2-Y81pCMF-GFP as shown in *Figure 12B and D*. At low surface protein concentrations [$c$(FGF2-Y81pCMF-GFP)≤0.4], FGF2-Y81pCMF-GFP clusters moved slightly slower in GUVs with membrane pores (D = 2.5 ± 0.8 μm$^2$/s) than in GUVs without membrane pores (D = 2.8 ± 0.5 μm$^2$/s). At high surface protein concentrations [$c$(FGF2-Y81pCMF-GFP)>0.4], the diffusion of FGF2-Y81pCMF-GFP membrane pore forming FGF2 oligomers was strongly decreased (D = 1.7 ± 0.4 μm$^2$/s). These data are consistent with the determination of oligomeric states shown in *Figure 12A and C*.

Finally, the methodology we used to determine oligomeric states and diffusion constants of FGF2-Y81pCMF-GFP on membrane surfaces was further verified using an FGF2 mutant that binds to PI(4,5)P$_2$ containing membranes, however, is incapable of oligomerizing and forming membrane pores [FGF2-Y81pCMF-C77/95A-GFP; *Figures 6* and *7*; (*Müller et al., 2015*)]. Indeed, the average oligomeric state of FGF2-Y81pCMF-C77/95A-GFP on membrane surfaces was determined to be 1.17 ± 0.5 along with a high diffusion constant of D = 4.8 ± 1.0 μm$^2$/s (*Figure 12E and F*). Consistently, GUVs containing membrane pores following incubation with FGF2-Y81pCMF-C77/95A-GFP were undetectable. These results demonstrate that this FGF2 variant form remained a monomer on the membrane surface of GUVs, which is consistent with previous studies and validates our experimental setup to determine the oligomeric state of FGF2-Y81pCMF-GFP on membrane surfaces. Thus, based on the data shown in *Figure 12*, FGF2-Y81pCMF-GFP oligomers in the membrane of GUVs are characterized by a range of about 8 to 12 subunits of FGF2 under conditions where membrane pore formation can be observed.

## Simultaneous interactions of FGF2 monomers with several PI(4,5)P$_2$ molecules stabilizes an FGF2 orientation that triggers dimerization

To complement experiments correlating the oligomeric state of FGF2 assemblies with membrane insertion and pore formation (*Figures 10*, *11* and *12*), we conducted atomistic molecular dynamics simulations to gain insight into the initial molecular events that trigger FGF2 oligomerization in a PI(4,5)P$_2$ dependent manner (*Figures 13* and *14*). A starting point of this approach were biochemical experiments demonstrating K127, R128 and K133 to be part of a binding pocket that recruits the headgroup of PI(4,5)P$_2$, IP$_3$ (17,33) (*Figure 6*). However, it is possible that additional residues play a role and the binding stoichiometry between FGF2 and PI(4,5)P$_2$ has not been determined.

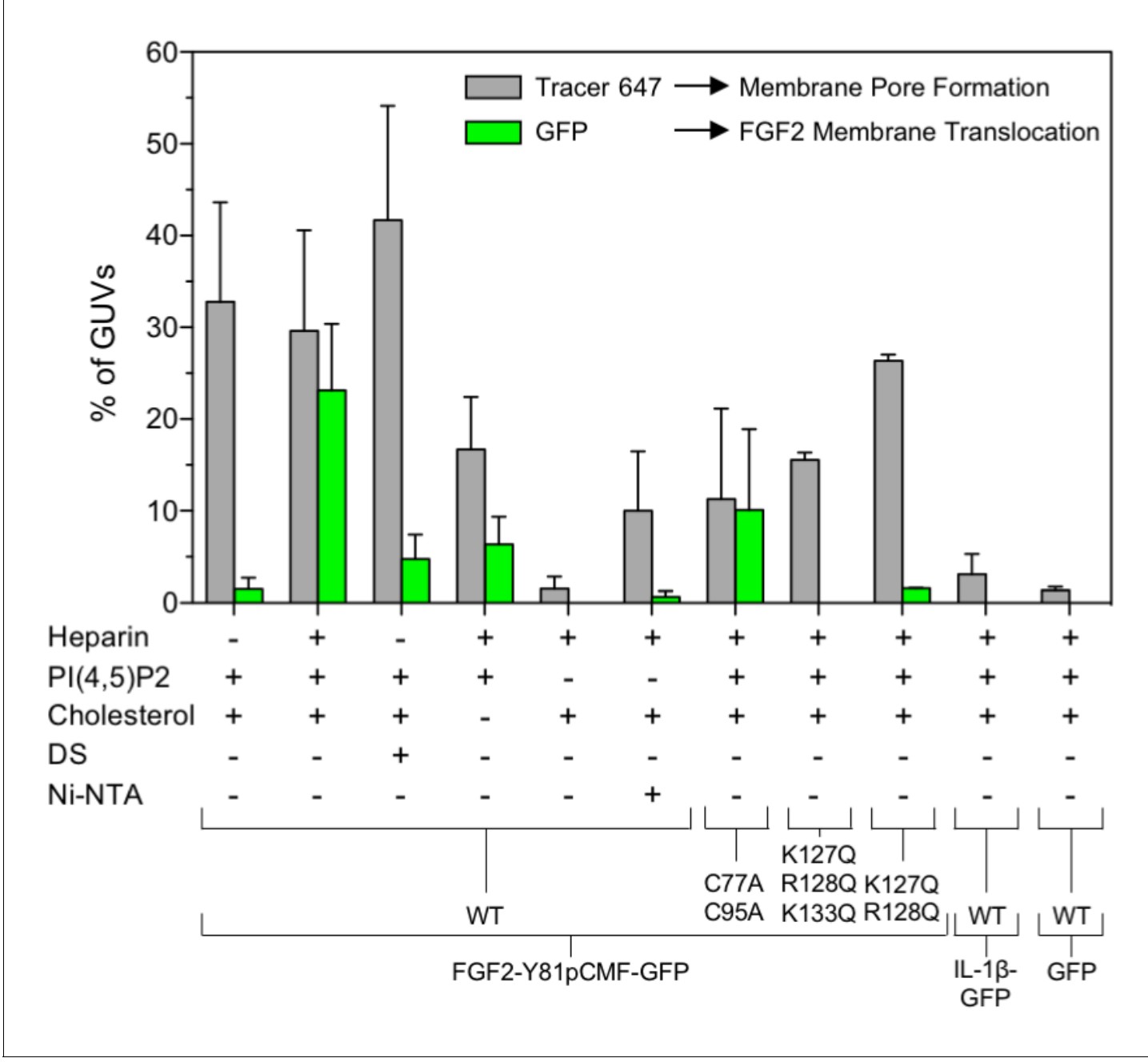

**Figure 9.** Quantification and statistical analysis of FGF2-Y81pCMF membrane translocation and its dependence on both *cis*-elements and *trans*-acting factors known to be required for FGF2 secretion from cells. A quantitative analysis of membrane translocation and pore formation by the various proteins indicated was conducted based upon the experiments shown in *Figures 3*, *4*, *5*, *7* and *8*. Various types of GUVs with a plasma membrane-like composition were used that differed with regard to the presence of the components indicated. For all conditions, data were derived from at least three independent experiments each of which involved the analysis of 20–120 GUVs per experimental condition. Gray bars indicate the percentage of GUVs with membrane pores with a ratio of Alexa647 tracer fluorescence in the lumen versus the exterior of $\geq 0.6$. Green bars indicate the percentage of GUVs where membrane translocation of GFP-tagged proteins had occurred with a ratio of GFP fluorescence in the lumen versus the exterior of $\geq 1.6$ being used as a threshold value. Standard deviations are shown ($n \geq 15$ for experiments shown in *Figure 3* and $n \geq 3$ for all other conditions shown in *Figures 4*, *5*, *7* and *8*). Detailed information on each individual experiment is provided in *Figure 9—source data 1*.

The following source data is available for figure 9:

**Source data 1.** Data for *Figure 9*.

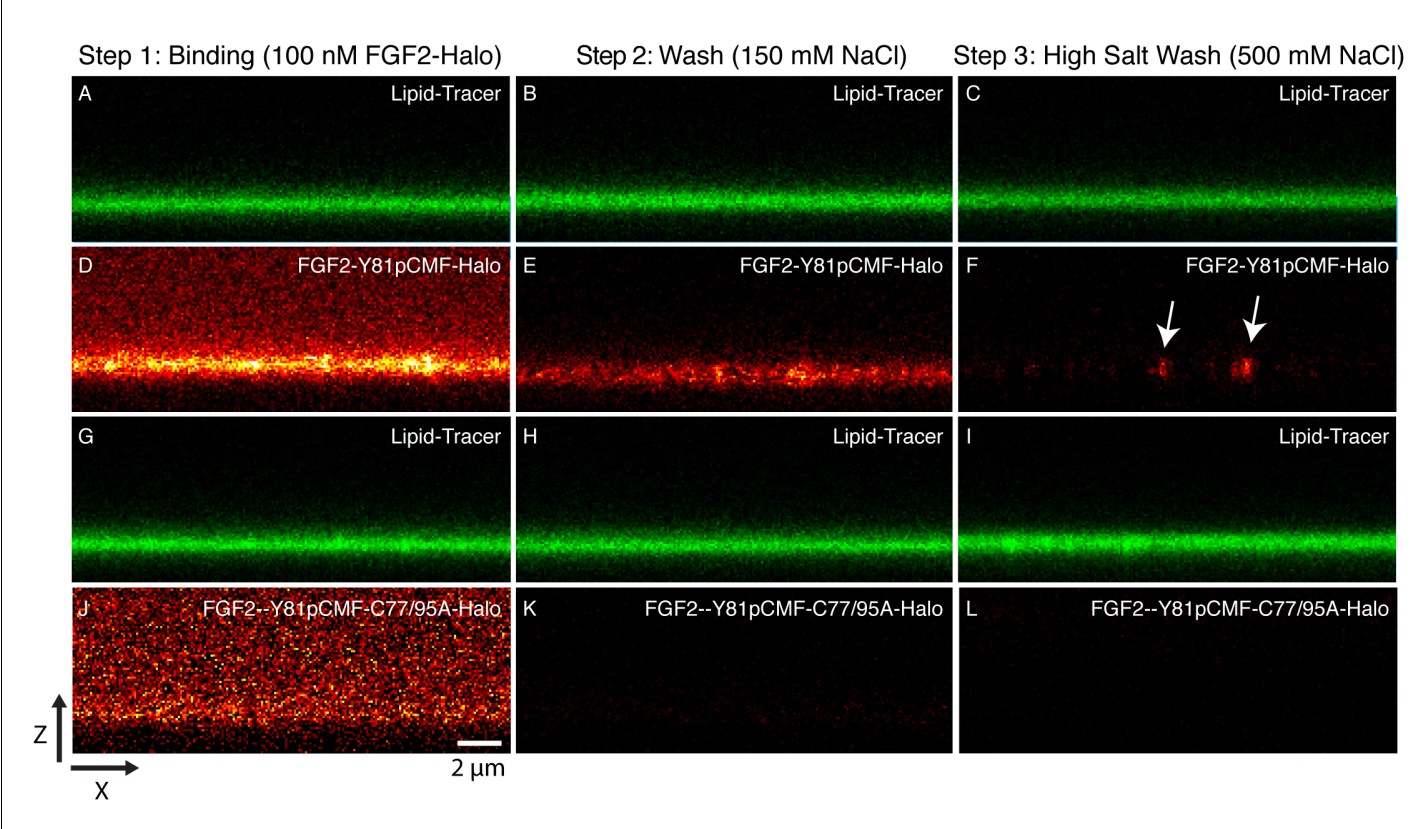

**Figure 10.** Binding and membrane insertion of FGF2-Halo-StarRed fusion proteins into supported lipid bilayers containing PI(4,5)P$_2$. FGF2-Y81pCMF-Halo-StarRed (Panels A–F) and FGF2-Y81pCMF-C77/95A-Halo-StarRed (panels G–L) were added at a final concentration of 100 nM to supported lipid bilayers (SLBs) containing 68 mol% POPC, 30 mol% cholesterol and 2 mol% PI(4,5)P$_2$ plus trace amounts of DPPE-OregonGreen to image the bilayer. FGF2-Y81pCMF-Halo-StarRed and FGF2-Y81pCMF-C77/95A-Halo-StarRed were bound to SLBs (panels A, D, G and J) followed by a 150 mM NaCl washing procedure (panels B, E, H and K). In a final step, a 500 mM salt wash was applied to remove FGF2 monomers (panels C, F, I and L). FGF2-Y81pCMF-Halo-StarRed and FGF2-Y81pCMF-C77/95A-Halo-StarRed bound to SLBs were imaged as explained in 'Materials and methods'.

Furthermore, beyond the formation of intermolecular disulfide bridges involving C77 and C95 (*Müller et al., 2015*), there is so far no information on protein-protein interfaces that form during PI(4,5)P$_2$ induced oligomerization of FGF2. Thus, we carried out a series of atomistic molecular dynamics simulations (*Figure 13* and *Video 1*; system M1 with five repeats) placing FGF2 monomers with a distance of 1.5 nm above the membrane surface in different orientations (*Figure 13* and *Figure 13—figure supplement 1*). The simulation data showed that FGF2 readily makes contacts with the membrane surface and undergoes changes in orientation as it binds to the membrane. In line with previous biochemical and structural studies (*La Venuta et al., 2015*; *Temmerman et al., 2008*; *Steringer et al., 2012*; *Müller et al., 2015*), we observed spontaneous binding of FGF2 to PI(4,5)P$_2$ through the key binding site residues (K127, R128, K133) (*Video 1*). However, as discussed below, a number of other residues were also found to contribute to the binding process depending on the number of PI(4,5)P$_2$ molecules interacting with FGF2 in its vicinity.

We further identified two different orientations of FGF2 on the membrane surface (*Figure 13*). The *high-affinity* orientation (*Figure 13A* and *Video 1*) observed in 3 out of 5 repeats is characterized by strong binding to the membrane surface. In this case, FGF2 orients in a manner where C95 is exposed in a way facilitating the formation of a C95–C95 disulfide bridge with a second monomer. In this orientation, C77 is not available to make a contact with a second FGF2 monomer. In this high-affinity orientation, all the three known binding site residues of FGF2 are involved in the interaction with PI(4,5)P$_2$ (*Figure 13*, *Figure 13—figure supplements 2* and *3*). The average binding energy for PI(4,5)P$_2$ is highest for K127 (–315.59 kJ/mol), followed by K133 (–301.25 kJ/mol) and R128 (–208.78

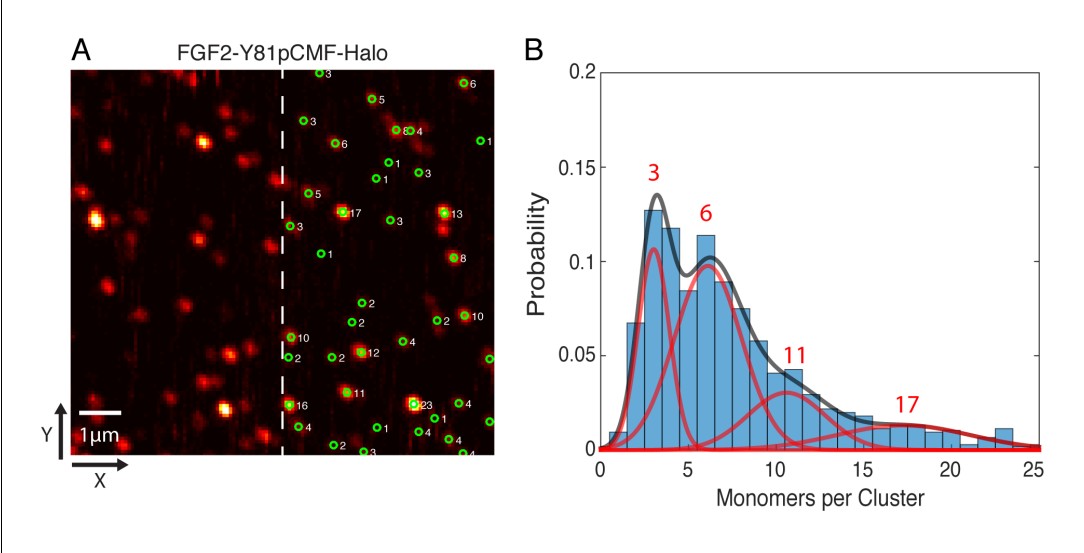

**Figure 11.** Single molecule imaging and brightness analysis of FGF2-Y81pCMF-Halo-StarRed to determine the oligomeric state of membrane inserted FGF2 clusters. (**A**) Immobile FGF2-Y81pCMF-Halo-StarRed clusters associated with SLBs following a high salt wash (**Figure 10F**) were imaged by confocal microscopy. The brightness of individual clusters was determined by fitting a 2D Gaussian to each diffraction limited spot using a single molecule tracking software in MATLAB. The number of monomers in each cluster (right part of image) was estimated by normalizing the brightness of each cluster to the brightness of monomeric HALO-StarRed). Original peak intensities and cluster analyses are available in **Figure 11—source data 1**. (**B**) Gaussian mixture analysis of the oligomeric state of membrane inserted FGF2-Y81pCMF-Halo-StarRed (>1000 clusters from six independent experiments). The distribution of monomers per cluster was complex. A Gaussian mixture analysis found 4 components with 3, 6, 11 and 17 monomers per cluster.

The following source data is available for figure 11:

**Source data 1.** Data for **Figure 11**.

kJ/mol). However, the interaction strengths were observed to depend on the number of PI(4,5)P$_2$ head groups interacting with the binding pocket region. In addition to the three key binding residues, a number of so far unidentified residues contributed to membrane binding: K34 (average interaction strength –113.65 kJ/mol), K137 (–244.72 kJ/mol) and K143 (–196.29 kJ/mol) (**Figure 13** and **Figure 13—figure supplement 4**). The number of head groups of PI(4,5)P$_2$ that were found bound to FGF2 varied between one and four with one to two head groups binding to the key residues in the primary binding site (K127, R128, K133) and one to two head groups bound to residues forming the second binding site (K34, K137, K143).

In the second *low-affinity* orientation (**Figure 13B**) that was observed in 2 out of 5 simulation repeats, the N- and C- terminal ends of FGF2 are close to the membrane surface, and both C95 and C77 are available to form disulfide bridges with other FGF2 monomers (**Figure 13B**). However, in this scenario, not all of the three key binding site residues interact with PI(4,5)P$_2$ as they point away from the PI(4,5)P$_2$ head groups. This is also evident from the average binding energies (K127 = –0.03 kJ/mol; R128 = –237.10 kJ/mol and K133 = –0.08 kJ/mol) (**Figure 13**, **Figure 13—figure supplements 5** and **6**), which are considerably weaker compared to the high-affinity orientation (**Figure 13** and **Figure 13—figure supplement 7**). Therefore, the weak interaction observed in the low-affinity orientation is likely to represent a transient binding intermediate at the membrane surface that is eventually stabilized when several PI(4,5)P$_2$ molecules are bound to FGF2.

Based on the above, the stability of FGF2-membrane binding is based partially on the FGF2 orientation, but also the number of PI(4,5)P$_2$ molecules that simultaneously bind to a FGF2 monomer is critical for stable membrane interactions of FGF2. This view is supported by additional simulations (system M2), where FGF2 was allowed to interact with only a single PI(4,5)P$_2$ molecule (see **Figure 13**, **Video 2**, **Figure 13—figure supplements 1B** and **8**). As expected, FGF2 was observed to bind to PI(4,5)P$_2$ with its key binding site residues, however during the course of the simulation the

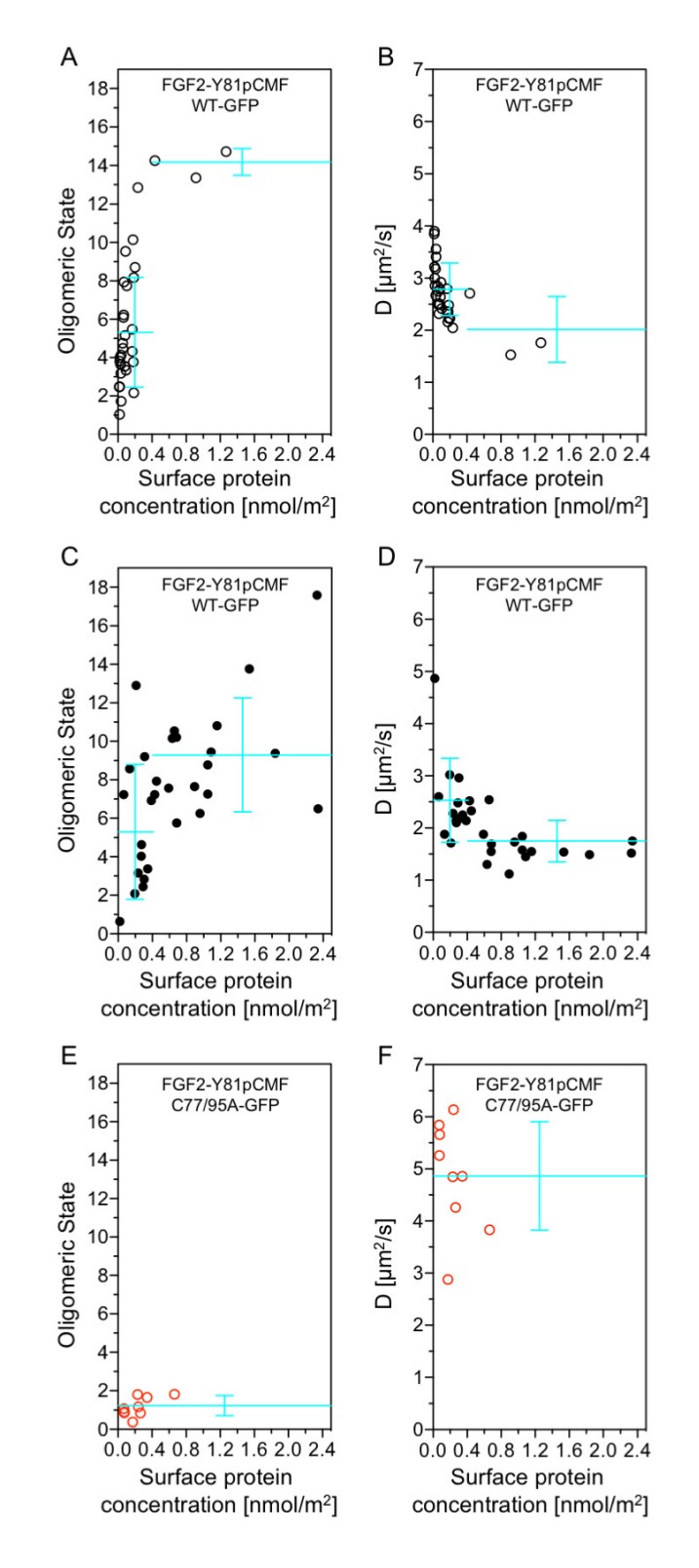

**Figure 12.** Functional correlation of the oligomeric state of membrane associated FGF2 translocation intermediates and membrane pore formation. Giant unilamellar vesicles with a plasma membrane-like lipid composition containing PI(4,5)P$_2$ and the membrane tracer DOPE-Atto633 were prepared as described in 'Materials and methods'. After pre-incubation with either the wild-type form of FGF2-Y81pCMF-GFP (panels A-D, black circles) or FGF2-Y81pCMF-C77/95A-GFP (panels E and F; red circles) for at least 30 min, z-scan FCS measurements using 515/50 nm (FGF2-GFP) and 697/58 nm (DOPE-

*Figure 12 continued*
Atto633) emission channels were conducted on single GUVs. The small free tracer AlexaFlour532 was added to the buffer in order to visualize FGF2 membrane pore formation. Accordingly, GUVs were classified into two groups with (panels **C** and **D**, filled circles) and without membrane pores (panels **A**, **B**, **E**, and **F**; empty circles). Z-scan measurements and analyses are described in detail under 'Materials and methods'. Average oligomeric state values (panels **A**, **C**, **E**) and diffusion constants (panels **B**, **D**, **F**) were plotted as a function of protein surface concentration. A total of 60 individual GUVs incubated with FGF2-Y81pCMF-WT-GFP (panels **A--D**) and 9 GUVs incubated with FGF2-Y81pCMF-C77/95A (panels **E** and **F**) were analyzed. Additional data of Z-scan FCS for each individual GUV, monomer control reference measurements as well as calculations of mean values with standard deviations are provided in *Figure 12—source data 1*.
The following source data is available for figure 12:

**Source data 1.** Data for *Figure 12*.

orientation of FGF2 fluctuated between high-affinity and low-affinity orientations. None of the two orientations was exceptionally stable as the inositol biphosphate ring (central in FGF2 binding) was too flexible to stabilize the structure of the complex. Thus, the simulations indicate that the surface area of FGF2 involved in binding to $PI(4,5)P_2$ is larger than previously assumed and renders interactions with more than one $PI(4,5)P_2$ molecule possible. This leads to strengthening the stability of FGF2 membrane binding suggesting that the high-affinity orientation is the most stable one.

## A model for $PI(4,5)P_2$ dependent FGF2 oligomerization based on C95-C95 disulfide linked dimers and C77-C77 disulfide bridges involved in the formation of higher FGF2 oligomers

The critical and most likely rate-limiting step of FGF2 oligomerization is $PI(4,5)P_2$ dependent dimerization. To identify possible dimerization interfaces along with the residues being involved, we conducted atomistic simulations (*Figure 14*). In previous FGF2 secretion assays and biochemical reconstitution experiments, FGF2 variant forms lacking both C95 and C77 were found defective in $PI(4,5)P_2$ dependent oligomerization, membrane pore formation and secretion from cells (*La Venuta et al., 2015*; *Müller et al., 2015*). When single cysteine substitutions were analyzed, the C95A variant form was characterized by a more severe defect in all of these assays compared to a C77A variant form of FGF2 (Müller, Wegehingel, Steringer and Nickel, unpublished results). To identify the residues involved in dimerization, we carried out two simulations of FGF2 trimers (system T) since they correspond to the minimal aggregation unit we identified in single particle brightness analyses using supported lipid bilayers (*Figures 10* and *11*). To this end, FGF2 monomers were placed in different orientations at a distance of 0.5 nm above the $PI(4,5)P_2$ head groups. The monomers were arranged such that two of them faced each other at a distance of 0.7 nm between their C95 residues. The third monomer resided 1.5 nm away from the other two monomers (*Figure 14* and *Figure 14—figure supplement 1*). In the first simulation, the two monomers readily oriented themselves into the high-affinity orientation and then dimerized. The dimer was observed to remain stable for the rest of the simulation. A detailed analysis revealed that there were four ion pairs formed across the interface (two pairs of E86 – K118 and E99 – K85 each; *Figure 14B*) that stabilized the dimer. At the same time, the third monomer diffused towards the dimer structure, however its orientation was closer to the low-affinity orientation than the high-affinity counterpart. Given this, it was quite expected that it bound to the dimer through an orientation where its C77 residue faced C77 in one of the FGF2 monomers in the dimer structure (*Figure 14C*). The binding across this C77 – C77 interface was based on the R47 – D49, D45 – R41, and R80 – E53 ion pairs that rendered the formation of this complex possible. Moving on, the second simulation of system T also highlighted that FGF2 monomers tend to aggregate, thus supporting the view that there is a force driving FGF2 monomers to form oligomeric structures. However, the time scale needed for orientation changes through rotational diffusion turned out to be long, thus no stable interface for dimerization was found during this type of simulation.

Biochemical and structural experiments have shown that disulfide bridges from C95 – C95 and C77 – C77 residue pairs are present in membrane inserted FGF2 oligomers ([*La Venuta et al., 2015*; *Müller et al., 2015*]; Müller and Nickel, unpublished results). Consistently, in our simulations, we observed formation of FGF2 dimers characterized by the C95 – C95 and C77 – C77 pairs at the

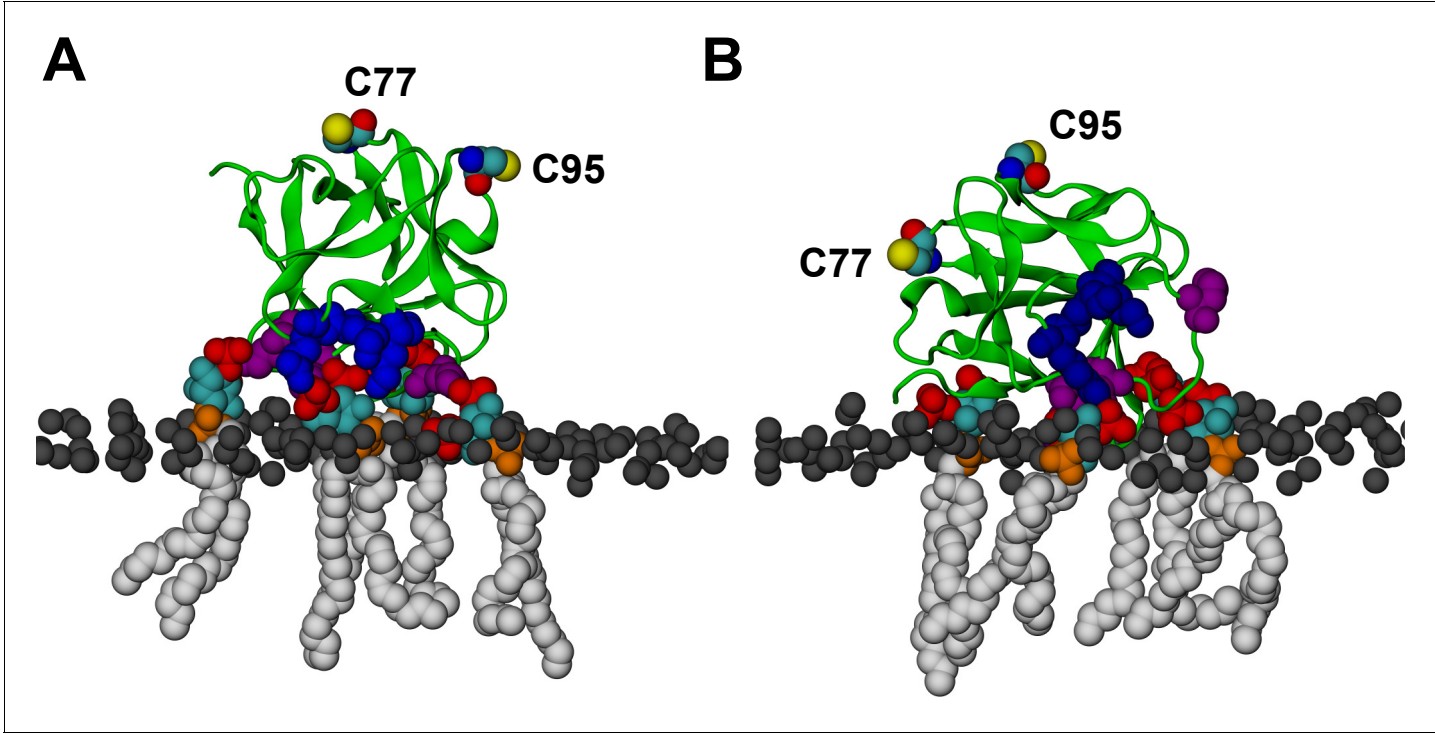

**Figure 13.** FGF2 orientation on the membrane surface. (**A**) High-affinity orientation of FGF2, showing all the known PI(4,5)P$_2$-binding site residues (K127, R128, K133) as well as the previously undetermined binding site residues (K34, K137, K143). (**B**) Low-affinity orientation of FGF2 in which the binding site residues lose contact with PI(4,5)P$_2$ and point away from the PI(4,5)P$_2$ head groups. FGF2 is rendered as green cartoon, and its C95 and C77 residues are shown as van der Waals (vdW) spheres and highlighted by text in the figure. The key binding pocket residues (K127, R128, K133) are shown as blue vdW spheres, and the additional binding site residues (K34, K137, K143) are shown as purple vdW spheres. Lipids are colored as gray vdW spheres (POPC phosphate atoms), red vdW spheres [PI(4,5)P$_2$ bisphosphates], cyan vdW spheres [inositol ring in PI(4,5)P$_2$], orange vdW spheres [phosphate linking the fatty acid chains and the inositol ring in PI(4,5)P$_2$], and white vdW spheres [fatty acid chains in PI(4,5)P$_2$]. Water molecules and ions are not shown for clarity.

The following figure supplements are available for figure 13:

**Figure supplement 1.** Initial structures in systems (A) M1 (POPC/cholesterol/PI(4,5)P$_2$ (65/29.5/5.5 composition on the cytosolic side interacting with FGF2); see text and *Table 1*) and (B) M2 (a single PI(4,5)P$_2$ molecule allowed to interact with FGF2).

**Figure supplement 2.** PI(4,5)P$_2$ binding energy based on electrostatics and van der Waals interactions.

**Figure supplement 3.** PI(4,5)P$_2$ interaction based on hydrogen bond analysis.

**Figure supplement 4.** PI(4,5)P$_2$ contacts with FGF2.

**Figure supplement 5.** PI(4,5)P$_2$ binding energy based on electrostatics and van der Waals interactions.

**Figure supplement 6.** PI(4,5)P$_2$ interaction based on hydrogen bond analysis.

**Figure supplement 7.** PI(4,5)P$_2$ contacts with FGF2.

**Figure supplement 8.** Binding of FGF2 to a single PI(4,5)P$_2$.

dimerization interface along with electrostatic interactions by additional residues in the dimer interface. Consequently, we used the dimer structure based on high-affinity FGF2 orientations with a stable C95 – C95 disulfide bridge in further simulations, where this link was established as a covalent bond to create a model of a full dimer (system D). The high-affinity orientation seen in simulations of

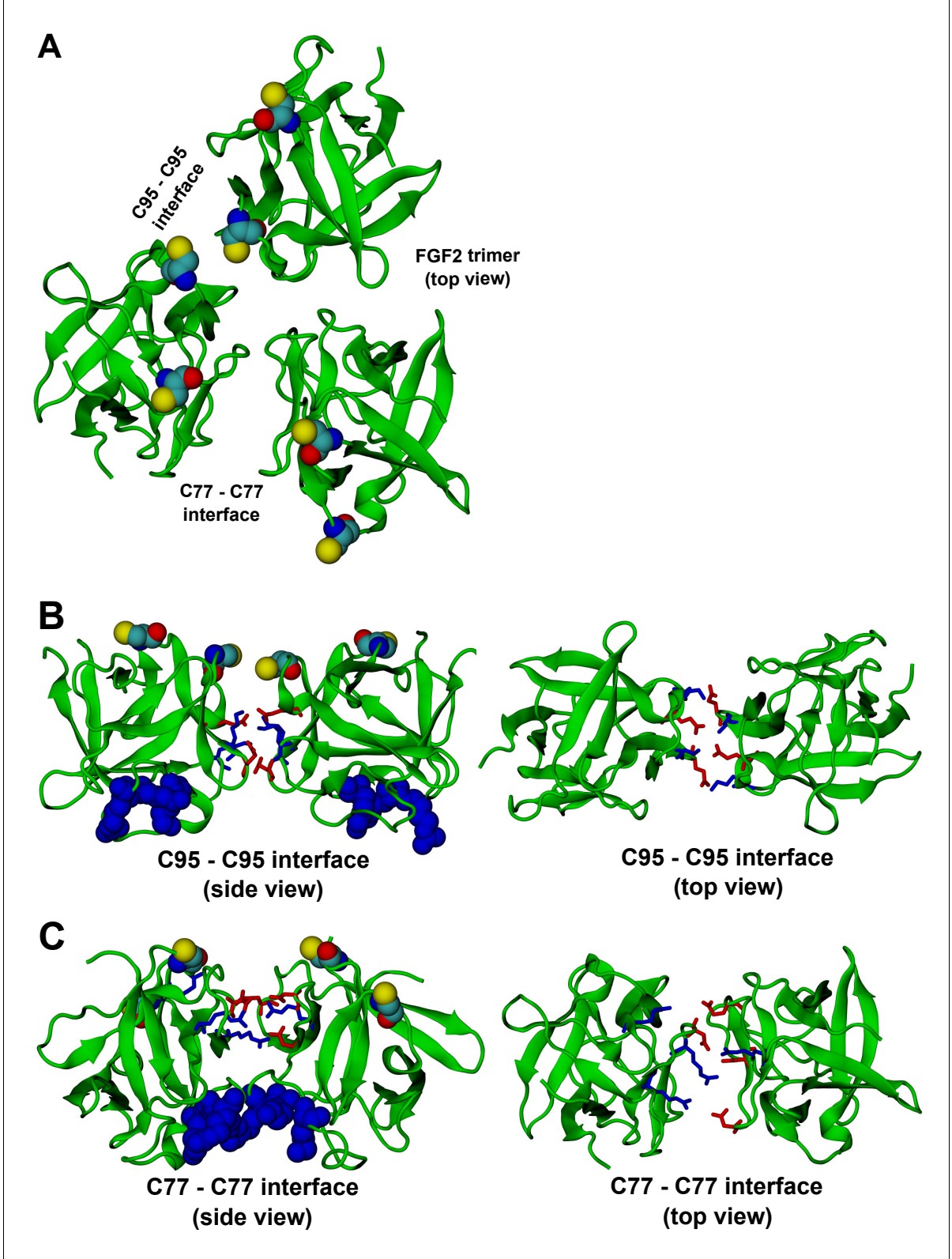

**Figure 14.** FGF2 trimer configurations. Snapshots representing the most populated structures in FGF2 trimer simulations (system T). (Panel **A**) depicts the top view of the FGF2 trimer aggregate with C95 – C95 and C77 – C77 interfaces labeled, where C95 and C77 are colored as van der Waals spheres. The trimer is split into two dimer interfaces shown in (panels **B** and **C**). (B) The interface residues involved in C95 – C95 disulfide-linked dimers. (C) The interface residues involved in C77 – C77 disulfide-linked dimers. The interface residues are depicted in stick representation, where negatively charged

*Figure 14 continued on next page*

*Figure 14 continued*

residues (D, E) are colored as red and positively charged residues (K, R) as blue. The PI(4,5)P$_2$ binding pocket residues (K127, R128, K133) are rendered as blue van der Waals spheres. For clarity, POPC, PI(4,5)P$_2$, cholesterol, water, and ions are not shown.

The following figure supplements are available for figure 14:

**Figure supplement 1.** FGF2 trimer in the beginning of the simulations.

**Figure supplement 2.** FGF2 dimer simulation.

**Figure supplement 3.** Dimerization interface.

system T was maintained during the simulations of system D. Importantly, with the dimer structure now fully stable due to the C95 – C95 covalent bond, the simulation of the dimer system D revealed several ion pairs (R80-D98; E86-K118 and R89-E99) (*Figure 14*, *Figure 14—figure supplements 2* and *3*). The extensive ion pairing formed at the interface as part of the high-affinity orientation suggests that the formation of FGF2 dimers containing the C95 – C95 disulfide bridge represents the initial step of PI(4,5)P$_2$ dependent FGF2 oligomerization and membrane pore formation.

## Discussion

The current study is the first of its kind in which an unconventional mechanism of protein secretion has been reconstituted from purified components establishing the molecular mechanism by which FGF2 is secreted from cells. We define the minimal machinery required for FGF2 membrane translocation and provide novel insights into the structure function relationship of membrane inserted FGF2 oligomers, the key intermediates of this process. Finally, using atomistic molecular dynamics simulations, this study puts forward a mechanism by which FGF2 monomers assemble into dimers and trimers on membrane surfaces in a PI(4,5)P$_2$ dependent manner, the rate limiting step for the formation of higher oligomers that form membrane pores.

The first part of this work provides direct proof for two critical predictions of a previously proposed model describing the molecular mechanism of unconventional secretion of FGF2 from cells (*La Venuta et al., 2015*; *Nickel, 2011*; *Nickel and Rabouille, 2009*, *2008*). The first predicted binding of FGF2 to PI(4,5)P$_2$ versus cell surface heparan sulfates to be mutually exclusive to ensure directional translocation of FGF2 across the plasma membrane based on sequential interactions of FGF2 with PI(4,5)P$_2$ at the inner leaflet and heparan sulfates at the outer leaflet. Here, using NMR spectroscopy we demonstrate an overlap of the binding sites on the molecular surface of FGF2 for IP$_3$ [the headgroup of PI(4,5)P$_2$] and a defined low affinity heparin disaccharide. In addition, we provide direct biochemical proof that long-chain

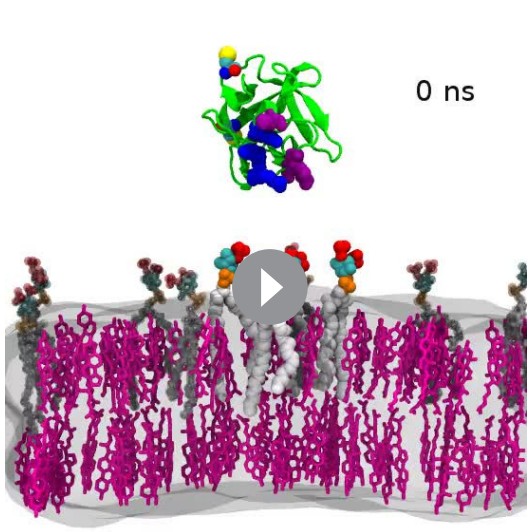

**Video 1.** FGF2 binding simultaneously to several PI(4,5)P$_2$ molecules (system M1). FGF2 binds to the membrane surface in the high-affinity orientation and interacts simultaneously with multiple PI(4,5)P$_2$, including the key residues in the main binding pocket (K127, R128, K133) and the residues in its vicinity (K34, K137, K143). The final frame of the video at 913 ns matches the snapshot in *Figure 13A* taken though from a slightly different perspective. The color coding used for the protein and PI(4,5)P$_2$ is consistent with *Figure 13*. In the membrane, cholesterols are shown in light purple, the phosphorous atoms of POPC molecules are depicted as a transparent surface, and for clarity's sake water and ions are not shown.

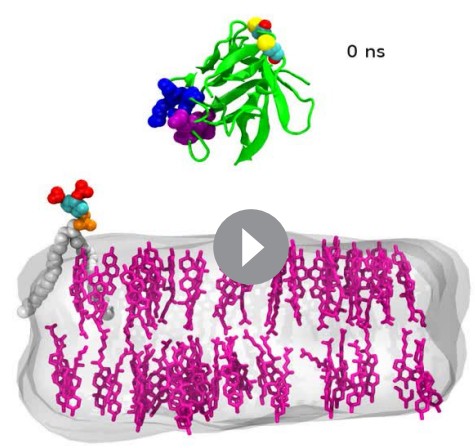

0 ns

**Video 2.** FGF2 binding to a single PI(4,5)P$_2$ (system M2). FGF2 adhering to the membrane surface through binding to a single PI(4,5)P$_2$. In the binding process, FGF2 first interacts with the residues close to the main binding site (K34, K137, K143, shown in purple van der Waals spheres), and then, at about 500 ns, rotates itself and interacts with the residues in the binding pocket (K127, R128, K133, shown in blue van der Waals spheres). The color coding used for the protein and PI(4,5)P$_2$ is consistent with *Figure 13*. In the membrane, cholesterols are shown in light purple, the phosphorous atoms of POPC molecules are depicted as a transparent surface, and for clarity's sake water and ions are not shown.

Legends for source data

heparins (used as mimetics of cell surface heparan sulfates) directly compete with membrane incorporated PI(4,5)P$_2$ for the interaction with FGF2. Based upon the high affinity of FGF2 towards heparan sulfates and long-chain heparins [K$_D$ ≈ 100 nM (*Faham et al., 1996*)] and the lower affinity towards PI(4,5)P$_2$ in the low micromolar range (*Temmerman et al., 2008*; *Temmerman and Nickel, 2009*) (*Figure 1*), our findings provide a direct explanation for vectorial translocation of FGF2 from the cytoplasm to the cell surface.

The second prediction of the FGF2 secretion model proposed the minimal machinery required for FGF2 membrane translocation to be composed of two *trans*-acting factors, PI(4,5)P$_2$ and cell surface heparan sulfates. This in turn suggests a direct requirement for *cis* elements in FGF2 that mediate binding to PI(4,5)P$_2$ (K127/R128) and heparan sulfates (K133). In addition, the core mechanism involves FGF2 oligomerization and membrane pore formation, a process that depends on two cysteine residues on the molecular surface of FGF2 (C77/C95). Using GUVs with a plasma membrane like lipid composition along with a FGF2-GFP fusion protein, we demonstrate FGF2 membrane translocation into the lumen of GUVs to depend on the presence of PI(4,5)P$_2$ on membrane surfaces and the presence of long-chain heparins in the lumen of GUVs. Beyond these *trans*-acting factors, we demonstrate *cis*-elements in FGF2 required for binding to PI(4,5)P$_2$ and heparin as well as for FGF2 oligomerization and membrane pore formation to be essential for FGF2 membrane translocation in a fully reconstituted system. By contrast, the *trans*-acting factor ATP1A1, even though required for FGF2 secretion from cells (*La Venuta et al., 2015*; *Zacherl et al., 2015*), apparently is dispensable for FGF2 membrane translocation under the conditions used in this study. This indicates that ATP1A1 has regulatory functions in FGF2 secretion from cells rather than belonging to the core machinery mediating physical translocation of FGF2 across the plasma membrane.

In the second part of this study, we used single molecule techniques to gain insight into the structure function relationship of membrane inserted FGF2 oligomers as key intermediates in membrane translocation. First, we used supported lipid bilayers (SLBs) and single molecule confocal microscopy to determine the oligomeric size distribution of membrane-inserted FGF2 species. Following short incubation times of 10 min during which the overall integrity of SLBs was fully maintained, three major oligomeric species could be detected represented by trimers, hexamers and higher oligomers in the range of about 10 to 12 subunits. These findings were corroborated by an independent single molecule system in which the average oligomeric state of FGF2 could be correlated with membrane insertion and pore formation in GUVs. Here, even after incubation times of several hours at high concentrations of FGF2, the average oligomeric state of FGF2 in GUVs leveled out in the range of 8 to 12 subunits. On the one hand, these experiments demonstrate membrane-inserted FGF2 oligomers to represent highly dynamic structures that are not characterized by a uniform number of subunits. On the other hand, our findings demonstrate that PI(4,5)P$_2$ dependent membrane recruitment does not result in the random formation of undefined FGF2 aggregates. Therefore, as discussed above, we propose that membrane inserted FGF2 oligomers serve as dynamic translocation intermediates by controlled assembly through PI(4,5)P$_2$ dependent FGF2 oligomerization at the inner leaflet and

controlled disassembly at the outer leaflet mediated by heparan sulfates. Previous evidence suggests that membrane inserted FGF2 oligomers are accommodated within a lipidic membrane pore with a toroidal architecture (*La Venuta et al., 2015*; *Steringer et al., 2012*). We propose that these translocation intermediates represent dynamic structures to which FGF2 subunits are constantly added at the cytoplasmic leaflet while FGF2 subunits are continuously removed at the extracellular side of the plasma membrane resulting in FGF2 translocation to the cell surface. It is currently unknown whether these assembly/disassembly units are monomers or disulfide bridged dimers of FGF2. Furthermore, it will be a challenge for future studies to elucidate the three-dimensional architecture of membrane inserted FGF2 oligomers to fully understand how these complexes are functioning as translocation intermediates in unconventional secretion of FGF2.

The third part of this study was concerned with the analysis of the initial events of FGF2 membrane translocation with a particular focus on the molecular mechanism of FGF2 interactions with PI(4,5)$P_2$ and the formation of dimers and trimers that initiate oligomerization of FGF2 in the context of PI(4,5)$P_2$ containing membranes. Atomistic simulations revealed the key binding site residues that interact with PI(4,5)$P_2$, which were found consistent with previous mutational analyses of FGF2 (6,17,19,25,33). Interestingly, the simulations predicted that there are additional previously unknown residues involved in membrane binding when several PI(4,5)$P_2$ head groups bind to FGF2 simultaneously. Therefore, the strength of FGF2 membrane interactions obviously depends on the number of PI(4,5)$P_2$ molecules that simultaenously bind to FGF2. In addition, the simulations provided compelling evidence that the strength of FGF2 membrane interactions also depends on the orientation of FGF2 relative to the membrane. The identified high-affinity orientation of FGF2 indeed favors multiple interactions of FGF2 with several PI(4,5)$P_2$ head groups and also leads to the formation of stable FGF2 dimers that are characterized by C95 – C95 disulfide bridges and additional ion pair interactions between residues within the dimerization interface. By contrast, the low-affinity orientation of FGF2 tilts the protein in a way that impairs simultaneous interactions of FGF2 with several PI(4,5)$P_2$ molecules and also prevents the formation of the C95 – C95 bridge. Given the critical role of C95 in FGF2 oligomerization, membrane pore formation, and secretion from cells, the simulations stress that FGF2 dimerization depends on membrane bound FGF2 monomers in the high-affinity orientation, whose stability is provided by multiple interactions with several PI(4,5)$P_2$ molecules. Therefore, the combined findings from these simulations predict that PI(4,5)$P_2$ dependent oligomerization of FGF2 is initiated through dimerization that is driven by the formation of C95 – C95 disulfide bridges. These FGF2 dimers appear to assemble into higher FGF2 oligomers driven by C77 – C77 disulfide links.

In conclusion, this study defines the minimal machinery required for FGF2 membrane translocation providing direct proof for a previously suggested model of unconventional secretion that was derived from cell-based data. This process depends on sequential and mutually exclusive interactions of FGF2 with PI(4,5)$P_2$ and heparan sulfates and is thermodynamically driven by FGF2 oligomerization and membrane insertion. These translocation intermediates are dynamic structures with a subunit number in the range of 8 to 12 FGF2 molecules. Therefore, depending only on two essential *trans*-acting factors, PI(4,5)$P_2$ and heparan sulfates, unconventional secretion of FGF2 is based upon a novel type of protein translocation across membranes with the cargo protein forming its own translocation intermediate by oligomerization and membrane insertion.

## Materials and methods

### Protein expression and purification

His-tagged variants of FGF2 (pQE30), FGF2-GFP and FGF2-Halo (both pET15b) were expressed in *E. coli* strains W3110Z1 or BL21 Star (DE3), respectively. For incorporation of the unnatural amino acid *p*-carboxylmethylphenylalanine (pCMF; custom synthesis by ENAMINE Ltd., Kiev, Ukraine), codon 81 (tyrosine) was replaced by an amber stop codon. Transformation of a strain carrying the pEVOL-pCMF plasmid resulted in expression of recombinant FGF2-Y81pCMF (*Young et al., 2010*). All proteins were purified in three steps via Ni-NTA affinity chromatography, heparin chromatography (except K127Q/R128Q and K127Q/R128Q/K133Q FGF2 variant forms) and size exclusion chromatography using a Superdex 75 column. In case of FGF2-Halo fusion proteins, desalting was performed using Nap-5 columns (GE Healthcare, Chicago, IL).

## NMR spectroscopy

Recombinant [15]N-labeled His-tagged FGF2-C77/95S (pQE30) was expressed in *Escherichia coli* W3110Z1 cells using M9 minimal medium with [15]NH$_4$Cl as the sole nitrogen source. Purification was performed as described above. Purified FGF2-C77/95S was diluted in 25 mM HEPES buffer (pH 7.4), containing 150 mM KCl and 10% D$_2$O, to a final concentration of 160 μM. For the IP$_3$ and heparin disaccharide titration experiments, 500 μl of 80 μM FGF2-C77/95S were titrated with defined volumes of 80 μM FGF2-C77/95S and either 900 μM IP$_3$ (Sigma 74148, Sigma Aldrich, St. Louis, MO) or heparin disaccharide (Sigma H9267). The endpoint of the IP$_3$ titration was further titrated with defined volumes of 80 μM FGF2-C77/95S and 900 μM heparin disaccharide and *vice versa*. NMR spectra were recorded on a Bruker AV 700 MHz NMR spectrometer equipped with a 5 mm triple resonance cryo-probe at 300 K. For each titration step 2D [15]N[1]H-HSQC spectra were recorded with 1024 points in the [1]H dimension and 96 points in the [15]N dimension and averaged over eight transients. Spectra were processed with TopSpin (Bruker, Billerica, MA) using CcpNmr Analysis software (*Vranken et al., 2005*). Signals were assigned using previously published data (accession code: 1BLA; (*Moy et al., 1996*, *1995*)) Chemical shift differences were calculated using the equation $\Delta\delta = \sqrt{(\delta H)^2 + (0,15 * \delta N)^2}$.

## Binding of FGF2 to PI(4,5)P$_2$-containing liposomes in the presence and absence of heparin

Large unilamellar vesicles (LUVs) with a plasma-membrane-like lipid composition either lacking (PM) or containing 2 mol% PI(4,5)P$_2$ (PM +PIP2) were prepared in buffer A (25 mM HEPES, pH 7.4, 150 mM KCl) supplemented with 10% (w/v) sucrose as described previously (*Steringer et al., 2012*). After blocking LUVs with 3% (w/v) fatty-acid free BSA in buffer A for 1 hr at 25°C, membranes were washed with buffer A and collected by sedimentation (15000x g; 20°C, 10 min). LUVs were resuspended in buffer A containing 2.5 μM His-FGF2-Y81pCMF-WT. After 1 hr of incubation at 25°C, either a mixture of long-chain heparins (Sigma H3149) or a defined heparin disaccharide (Sigma H9267) were added. In case of long-chain heparins, molar concentrations of FGF2 binding sites were defined by heparin disaccharide units with four sodium ions bound (MW 685). Following incubation at 25°C, unbound His-FGF2-Y81pCMF-WT and LUVs with bound His-FGF2-Y81pCMF-WT were separated by centrifugation (15000x g; 20°C, 10 min). While the supernatant (unbound material) was mixed with SDS-sample buffer, the sediment (bound material) was washed with buffer A followed by sedimentation of liposomes. The final pellet was resuspended in SDS-sample buffer. Samples (50% of bound and 14% of unbound material as well as 15% of input material) were analysed on 4–12% Bis-Tris SDS-PAGE in MES buffer (NuPAGE, Thermo Fisher Scientific, Waltham, MA). Proteins were stained with Coomassie Instant Blue (Expedeon, UK) followed by quantification and data normalization using the LI-COR Odyssey infrared imaging platform and Image Studio Lite Software (Version 5.2.5) (LICOR Biosciences, Lincoln, NE). FGF2 binding efficiency to PM-like liposomes containing PI(4,5)P$_2$ in the absence of heparin was set to 100% (bound material). Background binding was defined by PM-like liposomes lacking PI(4,5)P$_2$ (100% unbound material). Mean values with standard deviations (SD) are shown (n = 3).

## Characterization of FGF2 variant forms with regard to binding to heparin and PI(4,5)P$_2$

Heparin Sepharose 6 Fast Flow beads (20 μl; GE Healthcare) were equilibrated in 25 mM HEPES pH 7.4, 150 mM NaCl (buffer B). Heparin beads were incubated with the FGF2 variant forms indicated (2.5 μM protein concentration in 200 μl of buffer B) for 1 hr at room temperature on a rotating wheel. Afterwards, beads were pelleted (500 g; 25°C, 3 min) and the corresponding supernatants were treated with SDS sample buffer (unbound material). Heparin beads with bound proteins were washed three times with buffer B. Bound proteins were eluted with SDS-sample buffer. Input (5%), bound (20%), and unbound material (5%) were analyzed using 4–12% Bis-Tris acrylamide gels with MES buffer (NuPAGE, Thermo Fisher Scientific, Waltham, MA). Coomassie Instant Blue (Expedeon) stained bands were quantified on the LI-COR infrared imaging platform using Image Studio Lite Software (version 5.2.5). Mean values with standard deviations (SD) calculated from three independent experiments are shown.

FGF2 binding to PI(4,5)P$_2$ was quantified using flow cytometry as described previously (*Temmerman et al., 2008*; *Temmerman and Nickel, 2009*). PM-like LUVs (containing 1 mol% Rhodamine-PE) with or without 2 mol% PI(4,5)P$_2$ were blocked with 3% (w/v) fatty-acid free BSA in buffer A (25 mM HEPES, pH 7.4, 150 mM KCl) for 1 hr at 25°C. LUVs were sedimented, washed with buffer A and collected by centrifugation (16000x g; 25°C; 10 min). Following resuspension, LUVs were incubated with the various recombinant FGF2-GFP fusion proteins indicated at a final concentration of 2 µM in buffer A. Bound and unbound proteins were separated by sedimentation of LUVs followed by extensive washing in buffer A. The final liposome pellet was resuspended in 300 µl buffer A and analysed by flow cytometry using a FACS Calibur system (BD Biosciences, San Jose, CA). In brief, liposomes were gated by size and rhodamine fluorescence based upon the Rhodamine-PE membrane tracer. Simultaneously, GFP signals were measured to quantify binding of the various FGF2-GFP variant forms. To quantify protein-lipid interactions, raw data (fluorescent units) were corrected for liposome tethering using a shape index as reported previously (*Temmerman et al., 2008*; *Temmerman and Nickel, 2009*). Data were normalized defining GFP signals obtained with PM + PIP2 liposomes and FGF2-Y81pCMF-GFP-WT as 100% binding efficiency. Standard deviations are shown (n = 4).

## Preparation of giant unilamellar vesicles (GUVs)

GUVs with a plasma membrane like lipid composition consisting of 30 mol% cholesterol (Chol), 15 mol% sphingomyelin (SM), 34 mol% phosphatidylcholine (PC), 10 mol% phosphatidylethanolamine (PE), 5 mol% phosphatidylserine (PS), 5 mol% phosphatidylinositol (PI) and 1 mol% Biotinyl-PE (Avanti Polar Lipids, Alabaster, AL) were generated based on electro-swelling using platinum electrodes (*García-Sáez et al., 2009*). Where indicated, GUVs were supplemented with either PI(4,5)P$_2$ or a Ni-NTA lipid at 2 mol % at the expense of PC. In some experiments, GUVs were used lacking cholesterol which was supplemented with PC. For visualization either 0.05 mol% rhodamine B-labelled PE for FGF2 membrane translocation assays or 0.001 mol% DOPE-Atto 633 for z-scan FCS were added. The dried lipid film was hydrated with a 300 mM sucrose solution (300 mOsmol/kg). Where indicated, either long-chain heparins (50 µM; based on disaccharide units s.a.) or a defined heparin disaccharide (Sigma H9267) were included to mimic heparan sulfates in the lumen of GUVs. Osmolality was determined for all three conditions using a Wescor Vapro 5600 instrument and found not to be affected by the addition of long-chain heparins or the heparin disaccharide. Swelling was conducted at 45°C [10 Hz, 1.5 V for 50 min (without heparin) or 70 min (with heparin), 2 Hz, 1.5 V for 25 min]. In order to remove excess amounts of heparin, GUVs were gently washed with buffer B (25 mM HEPES pH7.4, 150 mM NaCl, 310 mOsmol/kg) and collected via centrifugation (1200x g; 25°C; 5 min). The loose GUV pellet was carefully resuspended in a small volume of buffer B and diluted again in 11.5 ml buffer B followed by centrifugation (1200x g; 25°C; 5 min). The supernatant was removed while the loose GUV pellet was carefully resuspended. Imaging chambers (LabTek) were incubated sequentially with 0.1 mg/ml Biotin-BSA (Sigma A8549) and 0.1 mg/ml Neutravidin (Thermo Fisher Scientific A2666) in buffer B. Luminal incorporation of heparin into GUVs was monitored by confocal microscopy in control experiments using a fluorescent derivative (Molecular probes; H7482). Likewise, the presence of PI(4,5)P$_2$ in the bilayers of GUVs was analyzed using a recombinant fusion protein of GFP with the PH domain of PLCδ1, a canonical PI(4,5)P$_2$ marker (*Lemmon, 2003*; *Milosevic et al., 2005*).

## Imaging and quantification of FGF2 membrane translocation using GUVs

For FGF2 membrane translocation assays, GUVs were incubated for 3 hr with a small fluorescent tracer (Alexa647) and the FGF2-GFP fusion proteins indicated at a final concentration of 200 nM. Confocal images were recorded at room temperature in multitrack mode using Zeiss LSM510 and LSM780 confocal fluorescence microscopes (Carl Zeiss AG, Oberkochen, Germany) along with a plan apochromat 63 x/1.4 oil immersion objective. Pinholes of the tracks were optimized to 1.2 µm. LSM510: In order to measure (i) GFP-, (ii) Rhodamine-PE-, and (iii) Alexa647-derived signals, samples were excited with (i) an argon laser (488 nm), (ii) a He-Ne-laser (561 nm), or (iii) a He-Ne laser (633 nm) and light was detected after (i) a band pass (BP) filter (505–530 nm), (ii) a BP filter (560–615 nm), or (iii) a long pass (LP) filter (>650 nm). Images were recorded in 8-bit grayscale. LSM780: Samples

were excited with (i) an argon laser (488 nm) for GFP, (ii) a He-Ne-laser (561 nm) for Rhodamine-PE, or (iii) a He-Ne laser (633 nm) for Alexa647 and light was split by a spectral beam guide to (i) 498–550 nm, (ii) 585–673, or (iii) 654–759 nm. Images were recorded in 12-bit grayscale and are shown pseudo-coloured in (i) green (Track2-ChS1), (ii) red (Track1-ChS2) and (iii) gray (Track2-Ch2). A radial profile analysis was conducted using ImageJ software (http://rsbweb.nih.gov/ij/) using the plugin 'radial profile' (http://rsbweb.nih.gov/ij/plugins/radialprofile.html) to detect intensity differences in the lumen, at the membrane, and in the surroundings of GUVs. Briefly, a circle was drawn around a GUV in a way that the center of the circle matched the center of the GUV in the confocal plane being assessed. From this center point the intensity at any given distance along the radius of the circle is measured and processed resulting in a profile plot of normalized integrated intensities around concentric circles as a function of distance from the center. Luminal fluorescence of individual GUVs was measured and normalized to fluorescence intensity of the surrounding buffer. Per experimental condition, 20 to 120 individual GUVs were analysed as indicated in the corresponding Fig. legends. To allow for a statistical analysis of membrane pore formation and FGF2-GFP membrane translocation across the population of GUVs, thresholds were defined to classify individual GUVs. When the ratio of inside to outside fluorescence of the small Alexa647 tracer was ≥0.6, GUVs were classified as vesicles containing membrane pores. Similarly, when the inside to outside ratio of GFP fluorescence was ≥1.6, the corresponding GUVs were classified as vesicles where FGF2-GFP membrane translocation into the lumen had occurred.

## 3D reconstruction of GUVs

3D reconstruction of GUVs (panels C and D of *Figure 3*) was based upon ImageJ software using the '3D Viewer' plugin (https://imagej.nih.gov/ij/plugins/3d-viewer/index.html). 3D images with a resolution of 512 × 512 pixels (0.07 μm pixel size) were made from z-stacks separated by a distance of 0.37 μm. All three channels – green (FGF2-Y81pCMF-GFP), red (Rhodamine-PE) and magenta (Alexa647) were merged into a single 3D image. Cross sections of GUVs are shown to visualize FGF2-Y81pCMF-GFP membrane translocation into the lumen.

## Analysis of FGF2 oligomerization on supported lipid bilayers

Protein labeling was conducted by incubating FGF2-Y81pCMF-Halo and FGF2-Y81pCMF-C77/95A-Halo, respectively, (20 μM each) with 40 μM StarRed-HTL in 150 mM NaCl, 20 mM HEPES (pH 7.4) for 1 hr at room temperature. Unbound dye was removed by size exclusion chromatography using a Nap-5 column (GE Healthcare). Degree of labelling (average number of dyes per FGF2 monomer) was determined to be DOL = 0.9 for both FGF2-Y81pCMF-Halo and FGF2-Y81pCMF-C77/95A-Halo, by measuring absorption spectra of the labelled protein and calculating the DOL according to:

$$DOL = \frac{A_{638} * \varepsilon_{Dye}}{(A_{280} - A_{280} * CF) * \varepsilon_{Prot}}$$

with $A$ being the local absorption maxima, $\varepsilon_{Dye} = 120000$, $CF = 0.32$, $\varepsilon_{Prot} = 79870$. Supported lipid bilayers (SLBs) were prepared on 18 mm round cover glasses. Glass surfaces were cleaned in 4% Hellmanex II in a bath sonicator for 10 min, rinsed and dried under nitrogen. Right before usage, cover glasses were plasma cleaned for 1 min. SLBs were formed by spin-coating at 3000 rpm for 1 min using 40 μl of lipid mixture [1.2 mg/ml; 68 mol% POPC, 30 mol% Cholesterol, 2 mol% PI(4,5)P$_2$ and 0.1 mol% DPPE-OregonGreen in methanol/chloroform (1:1)]. Residual solvent was evaporated for 1 hr under vacuum. Lipids were hydrated with a buffer containing 150 mM NaCl and 20 mM HEPES (pH 7.4). This procedure resulted in supported bilayers without defects. Occasionally, patches of double membranes were observed and avoided when studying FGF2 binding and oligomerization. FGF2-Y81pCMF-HALO-StarRed and FGF2-Y81pCMF-C77/95A-HALO-StarRed, respectively, were added to SLBs (final concentration = 100 nM) and incubated for 10 min. Membranes were washed in two steps with 150 mM NaCl buffer to remove unbound proteins and 500 mM NaCl to remove non-oligomerized FGF2 from membranes.

FGF2-HALO-StarRed fusion proteins associated with SLBs were imaged using a confocal laser scanning microscope with a 775 nm super-resolution STED module (Abberior Instruments, Göttingen, Germany). Fluorescence recovery after photo bleaching was performed by bleaching a 2 × 2 μm square with a 405 nm laser followed by measuring the diffusive recovery of FGF2 fusion proteins

over time. Diffusion constant and mobile fraction of FGF2 was estimated by fitting the recovery curves to a single exponential in MATLAB.

Single molecule imaging was done using 10 µW laser power (640 nm diode, 40 mMHz) in the backfocal plane of an 100x Oil NA 1.4 (Olympus) objective. Pixel size was 80 nm and pixel dwell-time 1 ms. To reduce background, we removed the first 0.5 ns of the detected photons by time-gating. This effectively reduced the background from scattered laser light. The resulting images were processed in MATLAB to extract the peak brightness per cluster using a 2D Gaussian fitting procedure. The number of monomers in each immobilized FGF2 cluster was estimated by normalizing the cluster brightness to the brightness of a single HALO-StarRed. To determine the brightness of a single HALO-StarRed, we expressed, purified and labelled mEGFP-HALO with StarRed-HTL. mEGFP-HALO-StarRed was then passively immobilized on clean cover glass such that single diffraction limited spots could be clearly separated. Imaging and single molecule brightness determination was identical with the procedure used for FGF2-HALO-StarRed fusion proteins. A Gaussian mixture analysis in MATLAB was conducted to estimate the number of sub-populations in the distribution of single molecule brightness and oligomeric number.

## Determination of average oligomeric states of GUV associated FGF2-Y81pCMF and FGF2-Y81pCMF-C77/95A-GFP by z-scan FCS

Measurements were performed on a self-constructed confocal microscope consisting of an inverted confocal microscope body IX71 (Olympus, Hamburg, Germany) and pulsed diode lasers (LDH-P-C-470, 470 nm, PicoTA, 532 nm, and LDH-D-C-635, 635 nm, PicoQuant, Berlin, Germany). The lasers were pulsing alternately at the repetition rate of 12.5 MHz to avoid artifacts caused by signal bleed-through. The laser light was coupled to a polarization maintaining single mode optical fiber and re-collimated at the output with an air space objective (UPLSAPO 4X, Olympus). The light was up-reflected to a water immersion objective (UPLSAPO 60x, Olympus) with a 465/533/635 dichroic mirror. The signal was split between two single photon avalanche diodes using 515/50 (FGF2-GFP) and 697/58 (DOPE-Atto 633) for FCS measurements or 515/50 (GFP) and 595/50 (small tracer Alexa-Fluor532) band pass filters (Chroma Rockingham, VT) to analyze membrane pore formation in GUVs selected for a FCS measurement. The laser intensity at the back aperture of the objective was kept below 10 µW for each laser line.

The determination of oligomeric states was based on a comparison of the brightness of a cluster $\phi_{\text{cluster}}$ to that one of a monomer $\phi_{\text{monomer}}$. GUVs were selected and classified with regard to the presence (small tracer AlexaFlour532 equilibrated between lumen and exterior) or absence (no tracer in the vesicle interior) of membrane pores. In the next step, a membrane was placed into the waist of 470 and 635 nm lasers, moved 1.5 µm below the waist and scanned vertically along the z axis in 20 steps spaced 150 nm from each other. FCS measurements (60 s each) using 515/50 nm (FGF2-GFP) and 697/58 nm (DOPE-Atto 633) emission channels were performed at each step. Auto-correlation (AC) curves corresponding to each of the 20 steps were fitted by a model assuming 2D diffusion within the membrane (bound FGF2-GFP/DOPE-Atto-633), 3D diffusion in solution (free FGF2-GFP) and transition of a dye to the triplet state (*Widengren et al., 1994*):

$$G(\tau) = 1 + \left( \frac{1}{PN_{\text{2D}}} \frac{1}{1 + (\tau/\tau_{\text{2D}})} + \frac{1}{PN_{\text{3D}}} \frac{1}{1 + (\tau/\tau_{\text{3D}})\sqrt{1 + SP(\tau/\tau_{\text{3D}})}} \right) \frac{1 - T + T\,\exp(-\tau/\tau_{\text{T}})}{1 - T} \qquad (1)$$

In *Equation (1)*, $\tau$ is the lag-time, $PN_x$ the particle number, $\tau_{\text{iD}}$ the dye diffusion time, $T$ the fraction of the dye in the triplet state and $\tau_{\text{T}}$ the lifetime of the triplet state. While auto correlation curves belonging to DOPE-Atto 633 were used to judge on quality of the lipid bilayer, those belonging to FGF2-GFP fusion proteins were used for the brightness analysis. Suitable properties of the membrane were indicated by free diffusion of DOPE-Atto 633 in the bilayer and a reasonable diffusion coefficient obtained for this probe. When plotting $PN_{\text{2D}}$(FGF2-GFP) against the vertical position z, a parabolic dependence is obtained due to the Gaussian shape of the excitation beam profile (*Benda et al., 2003*). The minimum value of $PN_{\text{2D}}(z, \text{FGF2-GFP})$, min[$PN_{\text{2D}}(z, \text{FGF2-GFP})$], obtained from such a plot and the corresponding average intensity in counts per second were used for further analysis. Position z corresponds to the middle of the membrane and allows for direct comparison of brightness values obtained for different measurements. Moreover, contribution from the bulk to the overall fluorescence signal is negligible at this position ($1/PN_{\text{3D}} \approx 0$). The brightness of a cluster was

calculated as $\phi_{\text{cluster}} = I/\min[PN_{\text{2D}}(z, \text{HisFGF2} - \text{Y81pCMF} - \text{GFP})]$. The brightness of a monomer $\phi_{\text{monomer}}$ was obtained in a similar manner as $\phi_{\text{cluster}}$, however, the probability that two labelled FGF2 molecules meet randomly in a cluster must be negligible. This has been achieved by (i) using the recombinant protein HisFGF2-Y81pCMF-C77/95A-GFP which binds to DGS-NTA containing bilayer as a dimer at maximum and (ii) diluting HisFGF2-Y81pCMF-C77/95A-GFP by the unlabeled variant HisFGF2-Y81pCMF-C77/95A. The average oligomeric state is calculated as

$$oligomeric\ state = \frac{\phi_{\text{cluster}}}{\phi_{\text{monomer}}} \tag{2}$$

The diffusion coefficient was obtained by plotting $\tau_{\text{2D}}$ against $z$ ($z$-scan FCS approach). In analogy to $PN_{\text{2D}}(z)$, a parabolic dependence was obtained. $D$ was determined by using the following relationship (**Benda et al., 2003**):

$$D = \frac{w_0^2}{4\min[\tau_{\text{2D}}(z)]} \tag{3}$$

with $w_0$ being the radius of the beam waist.

The protein surface concentration [$c(FGF2\text{-}GFP)$] was determined in two different ways. The first method makes use of the readouts that are obtained during the process of determining the oligomeric state of FGF2:

$$c(FGF2) = \frac{oligomeric\ state * PN_{\text{2D}}(\text{FGF2})}{\pi w_0^2} \tag{4}$$

The second method is based on the fact that fluorescence intensity is directly proportional to concentration. Therefore, by having a calibration solution with a given constant concentration of a dye, the actual concentration of FGF2-GFP at the membrane can be determined by the analysis of GUV fluorescence images. Both methods yielded similar results. The data shown in **Figure 12** were derived using the first method.

## Atomistic molecular dynamics simulations analyzing PI(4,5)P$_2$ dependent oligomerization of FGF2

For atomistic molecular dynamics (MD) simulations, we used the truncated FGF2 monomer structure [PDB id: 1BFF; (**Kastrup et al., 1997**)] from residue 26 to 154, without the flexible N-terminus. Cell based as well as biochemical experiments revealed that N-terminal truncations did not show defects in secretion nor oligomerization (André Engling, Julia Steringer, Hans-Michael Müller, Sebastian Unger and Walter Nickel, unpublished results). Here, we use the same residue numbering as in the PDB file (**Kastrup et al., 1997**), which is consistent with the numbering used in biochemical experiments. In preparation of the FGF2 structure for MD simulations, we processed the PDB structure with CHARMM-GUI (**Lee et al., 2016**), where we modeled all lysine (K) and arginine (R) residues as positively charged, glutamic acid (E) and aspartate (D) as negatively charged, histidine (H) and glutamine (Q) as uncharged, and tyrosine 81 was phosphorylated.

For the lipid bilayer models, we used a POPC/Cholesterol/PI(4,5)P$_2$ mixture (details below). PI(4,5)P$_2$ had 18:0 and 20:4 chains. The double bonds in the arachidonic fatty acid were located between the carbons C5 = C6, C8 = C9, C11 = C12, and C14 = C15. The overall charge in PI(4,5)P$_2$ was –4e. The bilayer systems prepared using CHARMM-GUI (**Lee et al., 2016**) were originally symmetric with respect to the transmembrane distribution of POPC and Cholesterol. PI(4,5)P$_2$ molecules were then inserted to the cytosolic leaflet of a membrane by removing a sufficient number of POPC molecules from the cytosolic leaflet and replacing the vacant space by PI(4,5)P$_2$, thus resulting in systems M1 and M2 (**Table 1**). System M1 had a POPC/Cholesterol (70/30) mixture in the extracellular leaflet and a POPC/Cholesterol/PI(4,5)P$_2$ (65/29.5/5.5) composition in the cytosolic monolayer. System M2 was designed to assess FGF2 binding in the dilute PI(4,5)P$_2$ limit, having only one PI(4,5)P$_2$ on the cytosolic side (**Table 1**).

In setting up the systems M1 (for five simulation repeats) with a single FGF2 monomer (**Table 1**), we placed the protein 1.5 nm above the membrane surface in five different orientations such that the key residues (K127, R128, K133) in the binding pocket for PI(4,5)P$_2$ either faced the PI(4,5)P$_2$ head groups or pointed away from them. For system M2, we placed a single FGF2 protein 1.5 nm

**Table 1.** Descriptions of model systems and their simulation details.

| System | No. of FGF2 molecules | No. of POPC molecules | No. of CHOL molecules | No. of PI(4,5)P2 molecules | No. of water molecules | Distance from membrane surface (nm) | No. of repeats | Simulation time (ns) |
|---|---|---|---|---|---|---|---|---|
| M1 | 1 | 348 | 152 | 14 | 34783 | 1.5 | 5 | 1000 × 5 |
| M2 | 1 | 177 | 76 | 1 | 15684 | 1.5 | 1 | 1000 |
| T | 3 | 348 | 152 | 14 | 31575 | 0.5 | 2 | 1000 × 2 |
| D | 2 | 348 | 152 | 14 | 29770 | * | 1 | 130 |

*The initial FGF2 dimer orientation and position on the membrane surface were the same as those in the final frame of the system T (repeat 2).

above the membrane surface with the protein's binding pocket residues facing the PI(4,5)P$_2$ head group.

The FGF2 trimer systems (T; see *Table 1*) simulated through two repeats were prepared by placing three FGF2 monomers 0.5 nm above the membrane surface with the binding pocket residues (K127, R128, K133) facing the PI(4,5)P$_2$ head groups, yet each of the monomers was positioned in a slightly different orientation with respect to the membrane surface. Here, the monomers were arranged such that two monomers faced each other with a C95 – C95 distance of 0.7 nm, and the third monomer was 1.5 nm away from the two other monomers.

For the FGF2 dimer system (D; see *Table 1*), the starting dimer structure was taken from the final frame of the trimer simulation (repeat 2) that demonstrated a stable dimer characterized by a C95 – C95 close-contact bridge with an upright orientation. We processed the dimer by creating a disulfide linkage between the two C95 residues by removing the hydrogen atom from the thiol groups. The C95 – C95 disulfide-linked dimer structure was energy minimized in vacuum with the membrane, thus not altering either the FGF2 dimer orientation or its interaction with the lipids.

The systems were neutralized by an appropriate number of potassium atoms. KCl salt at a concentration of 150 mM was added to mimic experimental conditions. Protein, lipids, and salt ions were described using the CHARMM36 force field (*Best et al., 2012*; *Klauda et al., 2010*). For water, we used the TIP3 model (*Jorgensen et al., 1983*).

All the systems were subjected to energy minimization using the steepest descent algorithm. After minimization, we ran 6 steps of equilibration runs where we gradually reduced the force constant applied to restrain the positions of the protein and the lipids. In the first stage of equilibration under NVT conditions, the Berendsen thermostat (*Berendsen et al., 1984*) was used to regulate the temperature at 298 K with a time constant of 1.0 ps. In the second stage of equilibration under NpT conditions, we used the Berendsen algorithm for controlling the simulation temperature and pressure along with a semi-isotropic pressure coupling scheme and a time constant set to 5.0 ps. In equilibration under NpT conditions, the reference pressure was set to 1.0 bar and the isothermal compressibility to a value of 4.5 × 10$^{-5}$ bar$^{-1}$. For neighbor searching, we used the Verlet scheme with a cut-off distance of 1.2 nm for short-range neighbor search. The electrostatic interactions were calculated using the PME method (*Darden et al., 1993*; *Essmann et al., 1995*). The cut-off length of 1.2 nm was used for both electrostatic (real space component) and van der Waals interactions. Hydrogen bonds were constrained using the LINCS algorithm (*Hess et al., 1997*) and periodic boundary conditions were applied in all directions. For the production MD runs, we removed all the restraints applied to the proteins and the lipids and used the Nose-Hoover thermostat (*Nosé, 1984*; *Hoover, 1985*) and Parrinello-Rahman barostat (*Parrinello and Rahman, 1981*) instead of the Berendsen's algorithm. The rest of the input parameters for production MD simulations were the same as those used under NpT conditions. The simulations were carried out using an integration time step of 2 fs with coordinates saved every 100 ps.

All simulations were performed using the GROMACS 5.1 (*Abraham et al., 2015*) simulation package and the analyses were done for the last 200 ns of the simulation trajectories using GROMACS tools and in-house scripts, except for system D, where the analysis was carried out over the last 30 ns. The images were rendered using VMD (*Humphrey et al., 1996*). The rendering was done for the starting structures and for the most populated structures obtained from clustering analysis based on RMSD values with a cut-off of 0.1 nm (*Daura et al., 1999*).

## Acknowledgements

The Abberior StarRed-HALO-Tag ligand was generously provided by Vladimir Belov (MPI-BPC, Göttingen, Germany). WN was supported by the German Research Foundation (DFG SFB/TRR 186, DFG SFB/TRR 83 and DFG Ni 423/6-1) and the DFG Cluster of Excellence at the University of Heidelberg. MH, SC and RS were supported by grant 14-03141 provided by the Czech Science Foundation and Praemium Academie award given by the Czech Academy of Sciences. CP, FL and IV were supported by the European Research Council (Advanced Grant 290974 CROWDED-PRO-LIPIDS), the Academy of Finland (Center of Excellence projects 272130 and 307415) and the Sigrid Juselius Foundation. CF was supported by the German Research Foundation as part of DFG SFB 854, DFG SFB 958 and DFG SFB/TRR 186. ÜC was supported by the DFG TRR83 TP18 grant and the German Federal Ministry of Education and Research grant to the German Center for Diabetes Research (DZD e.V.). IV and ÜC wish to acknowledge CSC – IT Center for Science (Espoo, Finland) and the Center for Information Services and High Performance Computing (ZIH (Technische Universität Dresden, Germany) for computational resources.

## Additional information

### Funding

| Funder | Grant reference number | Author |
|---|---|---|
| Czech Science Foundation | 14-03141 | Sabína Čujová<br>Radek Šachl<br>Martin Hof |
| European Research Council | Advanced Grant 290974 CROWDED-PRO-LIPIDS | Ilpo Vattulainen<br>Chetan Poojari<br>Fabio Lolicato |
| Academy of Finland | Center of Excellence project 272130 | Chetan Poojari<br>Fabio Lolicato<br>Ilpo Vattulainen |
| Sigrid Juselius Foundation | | Ilpo Vattulainen<br>Chetan Poojari<br>Fabio Lolicato |
| Academy of Finland | Center of Excellence project 307415 | Chetan Poojari<br>Fabio Lolicato<br>Ilpo Vattulainen |
| Deutsche Forschungsgemeinschaft | SFB/TRR 83 | Walter Nickel<br>Ünal Coskun |
| Deutsche Forschungsgemeinschaft | SFB 854 | Christian Freund |
| Deutsche Forschungsgemeinschaft | SFB 958 | Christian Freund |
| Deutsche Forschungsgemeinschaft | SFB/TRR 186 | Walter Nickel<br>Christian Freund |
| Deutsche Forschungsgemeinschaft | DFG Ni 423/6-1 | Walter Nickel |
| Bundesministerium für Bildung und Forschung | | Ünal Coskun |

The funders had no role in study design, data collection and interpretation, or the decision to submit the work for publication.

### Author contributions

JPS, Conceptualization, Data curation, Investigation; SL, Conceptualization, Data curation, Investigation, Methodology; SČ, CP, Data curation, Investigation, Methodology; RŠ, Conceptualization, Data curation, Formal analysis, Investigation, Methodology; FL, Data curation, Formal analysis, Investigation, Methodology; OB, SU, Data curation, Methodology; H-MM, Data curation, Methodology, Writing—review and editing; ÜC, Conceptualization, Resources, Writing—review and editing; AH,

Conceptualization, Data curation, Methodology, Writing—review and editing; IV, Conceptualization, Resources, Data curation, Methodology, Writing—review and editing; MH, Conceptualization, Resources, Methodology, Writing—review and editing; CF, Conceptualization, Data curation, Writing—review and editing; WN, Conceptualization, Formal analysis, Supervision, Funding acquisition, Writing—original draft, Project administration, Writing—review and editing

### Author ORCIDs

Julia P Steringer, http://orcid.org/0000-0001-9418-2762
Chetan Poojari, http://orcid.org/0000-0001-6575-221X
Fabio Lolicato, http://orcid.org/0000-0001-7537-0549
Sebastian Unger, http://orcid.org/0000-0003-2525-8502
Ünal Coskun, http://orcid.org/0000-0003-4375-3144
Ilpo Vattulainen, http://orcid.org/0000-0001-7408-3214
Walter Nickel, http://orcid.org/0000-0002-6496-8286

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
