## [Decision Letter]

[Editors’ note: a previous version of this study was rejected after peer review, but the authors submitted for reconsideration. The first decision letter after peer review is shown below.]

Thank you for submitting your work entitled "Minimal Machinery Required for Unconventional Secretion of Fibroblast Growth Factor 2" for consideration by *eLife*. Your article has been reviewed by three peer reviewers, one of whom is a member of our Board of Reviewing Editors, and the evaluation has been overseen by a Senior Editor. The reviewers have opted to remain anonymous.

Our decision has been reached after consultation between the reviewers. Based on these discussions and the individual reviews below, we regret to inform you that your work will not be considered further for publication in *eLife*.

As you will see from the reviewer's comments, the biochemical reconstitution of FGF2 translocation is considered a biochemical tour de force. Discussion amongst the reviewers, however, resulted in a unanimous view that, at present, the manuscript largely confirms the molecular requirements for FGF2 translocation that were identified previously. The reviewers and I encourage you to continue this line of experimentation, and to this end, they make several suggestions that may help you to provide new information regarding the mechanism of FGF2 translocation. Specific questions that the reviewers felt were especially important to address and expand on regard the nature of the FGF2 translocation intermediate and PIP2 dynamics. Specifically, does FGF2 form a structural pore (versus simply rupturing the membrane) and, related, does PIP2 translocate with FGF2, or remain on the 'donor' leaflet? Though the reviewers concluded that suitable revisions to the manuscript would likely substantial effort, they expressed enthusiasm for considering a manuscript that does contain additional information.

Reviewer #1:

This is a superbly conducted study that investigates the translocation of FGF2 across a synthetic membrane bilayer using a GUV-based experimental system. The authors find that binding of FGF2-GFP to PIP2 is required to initiate translocation and that lumenal heparin is required to accumulate FGF2-GFP within the lumen of the vesicle. Long chain heparin is shown to effectively compete for binding to PIP2, providing an explanation for the vectoral transport (i.e., secretion) previously observed with cells. The results generally support the conclusions of the manuscript, though I think that the authors need to address a couple of points to raise the significance of the manuscript.

1) I commend the authors for a rigorous and convincing reconstitution of FGF translocation, though the manuscript essentially confirms requirements defined by published work using cell-based approaches without revealing anything fundamentally new about the mechanism of translocation. Two suggestions for enhancing the significance of the work are listed below.

2) The conclusion that FGF2-GFP forms a pore, revealed by diffusion of Alexa 647 across the GUV membrane, has important implications regarding the mechanism of translocation. The formation of a toroid intermediate was alluded to (subsection “Biochemical reconstitution of FGF2 membrane translocation with purified components”), but the results don't distinguish the formation of a bona fide pore, which in my mind is constituted by a structure of defined stoichiometry and dimensions, versus a less definite breach of membrane integrity (e.g., due to crowding on the membrane) that results in leakage of the dye across the membrane. I wonder if a systematic test of soluble tracers of varying size might reveal a size limit that would lend support to the conclusion that a pore is an intermediate structure formed during translocation.

3) The conclusion that ATP1A1 plays no core role in FGF2 translocation is not satisfying because it raises the question, does the observed translocation truly reconstitute the physiological mechanism? I realize that this is technically challenging to address, but defining the contribution of ATP1A1, be it a core role or a regulatory function, would, (a) provide new information, and/or (b) might bolster the conclusion that it is not required for translocation.

Reviewer #2:

This is an interesting report that demonstrates a possible mechanism or at least a minimal system allowing translocation of FGF2 across a lipid bilayer based on FGF2 first binding and clustering on one side of a PIP-containing membrane and then flipping over to be captured by heparin on the other side. The in vitro results demonstrating feasibility are generally clear, and the mutational analysis, especially of mutants with perturbed binding to PIP2 but not heparin are consistent with the proposed cellular mechanism. Several issues should be addressed to strength the claim that this effect is what happens in the cell.

What is missing is discussion of how FGF2 traverses the hydrophobic part of the membrane or if it always travels with PIP2 and delvers it to the other leaflet. There is extensive literature about PIP2 flipping or enhancing flipping of other lipids by e.g. Sulpice and colleagues in platelets and other systems that seems highly relevant and should be cited and discussed in this context. E.g. Bucki R, Giraud F, Sulpice (2000) Phosphatidylinositol 4,5-bisphosphate domain inducers promote phospholipid transverse redistribution in biological membranes. Biochemistry 39: 5838-5844.

Is the only binding of FGF2 to PIP2 via the headgroup? If not, the competition between IP_3_ and heparin disaccharide lacks the full binding energy and it's not clear how to distinguish binding from electrostatic competition.

How specific is the effect of heparin? Is there a similar effect of hyaluronic acid, which might be more commonly bound on the outer cell surface?

How much of the interaction in vitro is due to electrostatics? How dependent is the binding affinity on ionic strength or divalent cations?

A control is needed to determine if heparin in the vesicle lumen changes the osmotic pressure inside and consequently the membrane tension.

A most interesting test would be between two vesicle types that have the same amount of PIP2 with heparin in the lumen but with different cholesterol content to shift from fully mixed to demixed bilayers, since this process is implicated in trafficking. The other controls are required technical controls but have no direct biological relevance.

Reviewer #3:

In their study, the authors reconstitute membrane translocation of FGF2 in vitro, confirming roles for PI(4,5)P_2_, heparan sulfates, and FGF2 oligomerization in the process. Other factors that promote FGF2 plasma membrane translocation in vivo were not required, suggesting they function only in a regulatory manner. Overall, the study is beautifully performed, and the data presented are rigorous and quantitative. However, the major concern is that the work is largely confirmatory in nature. Please see below.

1) Sequential interactions with PI(4,5)P_2_ and heparan sulfate was shown previously to be critical in FGF2 translocation across the plasma membrane. Also, the role for PI(4,5)P_2_ in promoting FGF2 oligomerization and membrane pore formation was also defined previously. The work done here solidifies/confirms our understanding of FGF2 translocation, but does not really extend it. In light of this, can the authors provide any evidence as to the mechanism by which ATP1A1 regulates FGF2 translocation, even if it is not absolutely required?

2) To extend the study, could the author investigate whether the addition of appropriately positioned cysteine residues to other members of the FGF family members, which have signal peptides, enables them to be transported across membranes in vitro?

[Editors’ note: what now follows is the decision letter after the authors resubmitted for further consideration.]

Thank you for resubmitting your work entitled "Key Steps in Unconventional Secretion of Fibroblast Growth Factor 2 Reconstituted with Purified Components" for further consideration at *eLife*. Your revised article has been favorably evaluated by Vivek Malhotra (Senior editor) and three reviewers, one of whom is a member of our Board of Reviewing Editors.

The manuscript has been improved and the reviewers recommend publication. You will see from the reviews below, which include the opinion of a new reviewer, that there are several points that should be addressed with additional revisions. These require only modification to the text suggested by reviewer 1 (point #3), reviewer # 2 (figure and the legend) and 3.

Reviewer #1:

The major issue raised during review of the original manuscript regarded the confirmatory nature of the manuscript, which confirmed specific requirements that were identified previously using cell based assays. In the revised manuscript, the authors now include several new findings that shed some light onto the mechanism of FGF2 translocation across a membrane. I find the most compelling of these to be the following:

1) Calibrated fluorescence measurements of labeled FGF2 was used to quantify the numbers of FGF molecules present in membrane-associated clusters. This provides some insight into possible mechanism of formation of an initial translocation intermediate.

2) Molecular dynamics simulations of FGF2 binding to PIP2, and FGF2 oligomerization, revealed possible new binding modes. Although these were not validated with 'wet' experiments, they provide some evidence to support the authors' speculation that FGF2 assembles into a larger pore-forming complex that mediates translocation.

3) The new data lead the authors to suggest a toroid translocation intermediate. I have a difficult time appreciating the manner in which FGF subunits might form a toroid intermediate, and especially how such a disulfide-bonded structure would be released from the cell, but the new data provide a foundation for this speculation. As this is not addressed in a substantial manner in the Discussion, the authors may wish to describe their thinking in a bit more detail.

Overall, my major concerns raised with the original submission have been substantially addressed.

Reviewer #2:

Fibroblast Growth Factors (FGFs) participate in a cell signaling pathways that are required for essential biological functions such as embryonic development, hematopoiesis, and wound repair. The focus of this manuscript is FGF2 which functions in normal cell growth and when mis-regulated plays a role in tumor-induced angiogenesis that is thought to be associated with tumor cell resistance to treatment therapies. FGF2 lacks a signal peptide and is transported into extracellular space by an ER/Golgi-independent mechanism. Results from cell-based experiments and in vitro studies, translocation of FGF2 is proposed to happen by direct protein translocation across the plasma membrane. This process depends on FGF2 oligomerization and sequential interaction with PI(4,5)P_2_ at the inner leaflet and heparan sulfate proteoglycans at the outer surface.

In this report the authors use NMR, in vitro translocation studies, and molecular dynamic simulations to study the mechanism of FGF2 translocation across membranes. Using NMR titration and binding assays the authors show that FGF2 binding to PI(4,5)P_2_ and long chain heparins are mutually exclusive and that FGF2 has higher binding affinity for heparan sulfates and long-chain heparins. Using GUVs the authors confirm that PI(4,5)P_2_ and heparan sulfates are sufficient to drive some FGF2 translocation across the membrane). They also confirm that FGF2 must oligomerize for translocation. Using supported lipid bilayers and single molecule techniques FGF2, the authors find that FGF2 oligomers are heterogeneous containing a range of ~8-12 subunits. Molecular dynamic simulations of early steps of FGF2 binding to PI(4,5)P_2_ confirmed that residues identified in previous mutational analysis studies are important for PI(4,5)P_2_ binding and that there may be previously unknown residues involved in PI(4,5)P_2_ binding when several PI(4,5)P_2_ head groups bind FGF2 at the same time.

The novelty of this manuscript is the establishment of an in vitro system that has allowed the authors to directly test the minimal components FGF2 translocation. However, the results appear to confirm previous finding about FGF2 translocation that were discovered using other approaches. While solid and important work, it may not be of high enough impact for publication in *eLife*. The paper would be significantly strengthened if the authors were able to show that their work has led to novel, or previously unknown mechanism of FGF2 translocation.

1) It would be helpful to see the entire HSQCs for the binding titrations. Are the residues that bind PI(4,5)P_2_ and Heparan the only ones that shift?

2) How did the authors confirm that the GUVs contained PI(4,5)P_2_ on their surfaces and contain long-chain heparins in their lumen?

3) How does the concentration of FGF2 used in the in vitro experiments compare to the amount of FGF2 present in cells. If the amount of FGF2 added to the GUVs is lowered does translocation still occur?

4) The last part of the paper present molecular dynamics studies that suggests that there are previously uncharacterized residues on FGF2 that are important for translocation. Were these tested in the in vitro system and in cells to confirm their importance?

*Reviewer #3:*

The revision of this manuscript makes important changes and adds significant new data involving the roles of cholesterol, the lack of effect of heparin disaccharide in translocation, and informative new single molecule studies. As a result this already strong report is further improved and will be of interest to a range of readers. Publication in its present form would be appropriate.

There are however, three points that might be considered. Not that they need to be fully resolved before acceptance of the work, but the issues might be worth taking into account for interpretation.

First, the inability of heparin disaccharides to mimic the effect of heparin might be interpreted differently. The result is analogous to a PIP2-binding protein that does not bind IP_3_ well. If the competition between PIP2 and heparin were analogous to an enzyme inhibitor and a substrate, a charge-matched heparin construct should compete for PIP2 if the concentration is enough. The need for a heparin polymer suggests that polyelectrolyte effects in addition to or instead of lock-key binding is relevant here. That was the reason to ask if HA can mimic heparin if it's packed in the vesicle lumen at high concentration. Heparin has nominally twice the charge density of HA, but maybe high concentration of HA would work. Without more data about e.g. the hydration state of the lipid and heparin bound to the FGF, it seems hard to distinguish traditional "specific" binding from equally specific charge density and polyelectrolyte effects.

Second, the experiments here are all done with only monovalent cations in the buffers. But in vivo, heparin is in a medium with mM Mg^2+^ and perhaps more important mM Ca^2+^. Most likely addition of extracellular concentration of Ca^2+^ or even intracellular Mg^2+^ makes the experiments intractable, but at least some mention of divalents and what they are known from literature to do to PIP2 and heparin would be useful.

Third, in principle some of the issues can be addressed by MD simulation, but the system here is too briefly described and possibly too limited to get into the atomic detail needed to get an accurate picture of the charge interactions. For example, when PIP2 or heparin bind FGF, are waters lost or retained at the interface of the negative oxygen of the ligand and the positive nitrogen of the protein? The fact that the net charge of PIP2 is –4 suggests that there are limits in the MD. Recent work from Radhakrishnan's group show that a fuller treatment of PIP2 under realistic conditions reduces the net charge closer to –3, which alters the way the lipid orients and binds ligands.

As mentioned above, none of these issues are probably easily resolvable and they need not delay this study. But perhaps addressing the issues would strengthen the interpretation.

---

## [Author Response]

[Editors’ note: the author responses to the first round of peer review follow.]

As you will see from the reviewer's comments, the biochemical reconstitution of FGF2 translocation is considered a biochemical tour de force. Discussion amongst the reviewers, however, resulted in a unanimous view that, at present, the manuscript largely confirms the molecular requirements for FGF2 translocation that were identified previously. The reviewers and I encourage you to continue this line of experimentation, and to this end, they make several suggestions that may help you to provide new information regarding the mechanism of FGF2 translocation. Specific questions that the reviewers felt were especially important to address and expand on regard the nature of the FGF2 translocation intermediate and PIP2 dynamics. Specifically, does FGF2 form a structural pore (versus simply rupturing the membrane) and, related, does PIP2 translocate with FGF2, or remain on the 'donor' leaflet? Though the reviewers concluded that suitable revisions to the manuscript would likely substantial effort, they expressed enthusiasm for considering a manuscript that does contain additional information.

We have submitted a revised version of the manuscript that contains both additional technical controls and two entirely new sections providing novel insights into the molecular mechanism of FGF2 membrane translocation. Beyond the reconstitution of FGF2 membrane translocation with purified components and the demonstration of mutually exclusive interactions of FGF2 with PI(4,5)P_2_ versus heparin, we have now added a series of single molecule studies that functionally correlate the oligomeric state of PI(4,5)P_2_ induced FGF2 oligomers with membrane insertion and pore formation. Furthermore, the manuscript now contains a series of atomistic molecular dynamics simulations that provide novel insights into the binding mechanism of FGF2 to PI(4,5)P_2_ as well as into the molecular mechanism of the initial steps of FGF2 oligomerization triggered by PI(4,5)P_2_. With these modifications, the revised manuscript addresses the three major aspects that were raised by the reviewers:

1) We have added additional controls to the core part of the original manuscript. This concerns experiments addressing questions of specificity when different forms of heparin are used as luminal traps for FGF2 in GUVs (new Figure 4 and Figure 9) as well as a potential role of cholesterol in PI(4,5)P_2_ dependent FGF2 membrane translocation (new Figure 5 and Figure 9).

2) We have added an entirely new section providing a functional single molecule

analysis of membrane-inserted FGF2 oligomers (new Figure 10, Figure 11 and Figure 12). Following the suggestions of the reviewers, these studies provide the first insights into the structure-function relationship of membrane-inserted FGF2 oligomers. Our findings demonstrate PI(4,5)P_2_ dependent FGF2 oligomers to represent dynamic structures in the range of 10 to 12 subunits that form membrane-inserted translocation intermediates during unconventional secretion of FGF2. These findings are discussed in the context of a toroidal membrane pore and the assembly/disassembly hypothesis of FGF2 membrane translocation we have put forward previously [reviewed in (1)].

3) We added a new section based upon atomistic molecular dynamics simulations providing clues about the molecular mechanism of FGF2 binding to PI(4,5)P_2_ and the initial steps of PI(4,5)P_2_ induced oligomerization of FGF2 (new Figure 13 and Figure 14, Figure 13—figure supplement 1-8 and Figure 14—figure supplement 1-3 and Video 1 and Video 2). Key aspects of this part are (i) the identification of additional binding sites in FGF2 resulting in multiple interactions of one FGF2 molecule with several PI(4,5)P_2_ molecules and (ii) the identification of a high affinity dimer interface in FGF2 highlighting a disulfide bridge connecting two cysteine 95 residues. These results are discussed in the context of the FGF2 membrane translocation model we are putting forward with the current manuscript.

Reviewer #1:

1.) Oligomeric state of membrane-inserted FGF2 forms and size limit of membrane

pores generated by PI(4,5)P_2_ induced FGF2 oligomers

In previous studies, we have conducted experiments analyzing a potential size cutoff of membrane pores generated by PI(4,5)P_2_ dependent FGF2 oligomerization and membrane insertion (2). While we found an Alexa488 dye (1 kDa) to physically traverse the membrane, the same dye coupled to either a dextran molecule (10 kDa) or cytochrome c (12 kDa) did not. Furthermore, we have previously shown that, upon membrane insertion of FGF2 oligomers, transbilayer diffusion of membrane lipids occurs providing direct evidence for the formation of a toroidal pore (2). These findings suggest that FGF2 oligomers insert into membranes through electrostatic interactions with PI(4,5)P_2_ thereby opening up a lipidic pore with a toroidal structure. In the revised manuscript, we have now included a series of single molecule measurements using both GUVs and supported lipid bilayers functionally correlating the oligomeric state of FGF2 with membrane pore formation (new Figure 10, Figure 11 and Figure 12). We find membrane-inserted FGF2 oligomers to represent dynamic structures with pore forming species characterized by about 10 to 12 FGF2 subunits. As discussed in the revised manuscript, these findings are consistent with a toroidal architecture and point to a dynamic structure that is continuously assembled at the inner leaflet and disassembled at the outer leaflet resulting in net transport of FGF2 molecules across the membrane.

*2) The role of ATP1A1 in FGF2 membrane translocation*

The fact that FGF2 membrane translocation can be observed in the absence of ATP1A1 does not imply that our reconstitution system is unrelated to the physiological mechanism. The most simple explanation would be that ATP1A1 is required in cells as a first physical contact to accumulate FGF2 at the inner leaflet of the plasma membrane at sites that provide proximity to Tec kinase and/or PI(4,5)P_2_. This requirement might be simply bypassed in the reconstitution system when the essential components are present in a purified form. However, provided ATP1A1 could be included in a proper way into our reconstitution system, it is possible that the efficiency of FGF2 membrane translocation would increase. Therefore, while ATP1A1 clearly is not essential for FGF2 membrane translocation to occur, we are indeed investigating ways of reconstituting ATP1A1 as part of our FGF2 membrane translocation assay. However, as noted by reviewer #1, reconstituting multi-subunit integral membrane proteins in a functional form is technically challenging and beyond the focus of the current manuscript.

*Reviewer #2:*

*[…] 1) How does FGF2 traverse the hydrophobic part of the membrane?*

The results from various studies point at a toroidal architecture of FGF2 induced membrane pores where the FGF2 oligomer is located in the center with each monomer bound to PI(4,5)P_2_ [reviewed in (1)]. The combined findings led to the assembly/disassembly model of FGF2 membrane translocation where FGF2 subunits are added to membrane-inserted FGF2 oligomers at the inner leaflet followed by removal of FGF2 subunits at the outer leaflet. The latter step is mediated by heparan sulfates that outcompete PI(4,5)P_2_ with regard to binding to FGF2. This process results in directional transport across the plasma membrane along with FGF2 trapping in the extracellular space where FGF2 remains bound to heparan sulfates on cell surfaces. As also explained in our response to reviewer #1, this view is supported by several lines of evidence. First, a toroidal architecture of FGF2 membrane pores is indicated by transbilayer diffusion of membrane lipids observed following membrane insertion of FGF2 oligomers (2). These results were corroborated by findings demonstrating that FGF2 oligomers can be extracted from membranes using carbonate, a direct indication of electrostatic rather than hydrophobic interactions with the membrane (3). Finally, the assembly/disassembly model as part of a toroidal pore is strongly supported by the current study demonstrating mutually exclusive interactions of FGF2 with PI(4,5)P_2_ and heparin, the latter mimicking cellular heparan sulfates outcompeting PI(4,5)P_2_ and thereby trapping FGF2 at the extracellular leaflet. In conclusion, all data from various studies point to a mechanism of FGF2 membrane translocation that is characterized by a lack of physical contacts of FGF2 with the hydrophobic core of the membrane. With regard to transbilayer diffusion of membrane lipids (2), it is indeed likely that this phenomenon includes PI(4,5)P_2_ itself as part of the assembly/disassembly mechanism of FGF2 membrane translocation. As suggested by reviewer #2, in the revised manuscript these aspects are now discussed in more detail along with the relevant literature on domain induced PIP2 flipping in cellular membranes.

2) Does FGF2 exclusively bind to the headgroup of PI(4,5)P2 ?

There is no evidence that FGF2 interacts with the hydrophobic parts of PI(4,5)P_2_. Using atomistic molecular dynamics simulations, we have further tested this possibility and, consistent with previous biochemical and structural data, found a high affinity binding site for the headgroup of PI(4,5)P_2_. (new Figure 13 and Figure 14, Figure 13—figure supplement 1-8 and Figure 14—figure supplement 1-3 and Video 1 and Video 2). Intriguingly, these studies revealed additional binding sites in FGF2 for PI(4,5)P_2_, suggesting that FGF2 accumulates PI(4,5)P_2_ molecules at the sites of oligomerization and membrane pore formation. These findings are of extraordinary relevance for the formation of membrane pores with a toroidal architecture as we have shown previously that PI(4,5)P_2_ is required for FGF2 membrane insertion based upon its ability to stabilize the high membrane curvature that characterizes toroidal pores (2). The molecular dynamics studies included into the revised manuscript further provide clues on the molecular mechanism of the initial steps of PI(4,5)P_2_ induced FGF2 oligomerization which will be highly relevant for future studies analyzing the fine structure of FGF2 oligomers as part of toroidal pores.

3) How specific is the effect of heparin? Could it be replaced by for example hyarulonic acid?

FGF family members are well characterized with regard to their ligand binding specificities. FGF2 has been shown to interact exclusively with sulphated glycosaminoglycans as contained in heparan sulfate proteoglycans. As opposed to heparan sulfates and other sulfated proteoglycans, cellular forms of hyaluronic acid do not contain sulfate groups. In any case, reviewer #2 inspired us to systematically compare long-chain heparins with a defined heparin disaccharide (as used in Figure 1 and Figure 2) in FGF2 membrane translocation assays. As shown in the new Figure 4 and quantified in the new Figure 9, replacing long-chain heparins by a heparin disaccharide does not have any impact on membrane pore formation, however, FGF2 membrane translocation is almost fully blocked. These findings are in line with the NMR data and biochemical experiments shown in Figure 1 and Figure 2 demonstrating long-chain heparins to be required to outcompete PI(4,5)P_2_ with regard to binding to FGF2. Our combined results therefore corroborate the proposed model for the molecular mechanism of directional membrane translocation of FGF2 based on sequential interactions with PI(4,5)P_2_ and heparan sufates on opposing sides of the membrane.

4) Does heparin change the osmotic pressure and, in turn, membrane tension?

As explained in “Materials and methods”, the generation procedures of all types and batches of GUVs were controlled by osmolality measurements and demonstrated not to be affected by adding either long-chain or disaccharide heparins. Therefore, artifacts related to differences in osmotic pressure and membrane tension can be excluded.

5) Does cholesterol play a role in FGF2 membrane translocation?

As suggested by reviewer #2, we have conducted reconstitution experiments with heparin-containing GUVs that either contained or lacked cholesterol. As shown in the new Figure 5 and quantified in the new Figure 9, omission of cholesterol causes a substantial decrease of both membrane pore formation and translocation of FGF2. The experiments shown in Figure 5 indicate that, in the absence of cholesterol, binding of FGF2 to PI(4,5)P_2_ is strongly impaired. This is consistent with previous studies from our laboratory demonstrating strong binding of FGF2 to PI(4,5)P_2_ in liposomes with a plasma membrane like lipid composition whereas binding was found poor when PI(4,5)P_2_ was reconstituted in pure PC liposomes (4,5). Our combined findings suggest that components of the FGF2 secretion apparatus such as PI(4,5)P_2_ are organized in cholesterol dependent microdomains that are required for efficient recruitment, oligomerization and membrane translocation of FGF2. The new results are shown in Figure 5 and Figure 9 in the revised manuscript and are discussed accordingly.

*Reviewer #3:*

1) What could be the role of ATP1A1 as a novel aspect of FGF2 membrane translocation?

In previous studies, we have demonstrated a direct interaction of FGF2 with the cytoplasmic domain of ATP1A1 (6). As explained in our response to reviewer #1, the function of ATP1A1 in FGF2 secretion from cells might be a first physical contact of FGF2 with the inner leaflet of plasma membranes. One possible explanation for a requirement of ATP1A1 in FGF2 secretion from cells might be a need to accumulate FGF2 in plasma membrane microdomains mediating proximity to additional components such as Tec kinase and PI(4,5)P_2_. In our reconstitution system, this function might be bypassed when the essential components are present in a purified form. While we do run projects analyzing the function of ATP1A1 both in a cellular context and in reconstitution systems, these studies are clearly beyond the focus of the current manuscript.

2) Do cysteine residues transplanted to FGF family members with signal peptides promote membrane translocation similar to FGF_2_?

We have done such experiments in the context of cells (7). As a model FGF protein carrying a signal peptide, FGF4 has been used as it is a close structural relative of FGF2. Following removal of the signal peptide of FGF4, secretion became undetectable, however, after transplanting the critical cis-elements including Cys77 and Cys95, the corresponding FGF4/2 hybrid protein was indeed found rerouted into the unconventional secretory pathway of FGF2 (7). Accordingly, a double cysteine mutant of FGF2 (C77/95A) was used in the current study and demonstrated to be strongly impaired in membrane translocation (Figure 7 and Figure 9).

References:

1. La Venuta, G., Zeitler, M., Steringer, J. P., Müller, H. M., and Nickel, W. (2015) The Startling Properties of Fibroblast Growth Factor 2: How to Exit Mammalian Cells without a Signal Peptide at Hand. J Biol Chem 290, 27015-27020.

2. Steringer, J. P., Bleicken, S., Andreas, H., Zacherl, S., Laussmann, M., Temmerman, K., Contreras, F. X., Bharat, T. A., Lechner, J., Müller, H. M., Briggs, J. A., Garcia-Saez, A. J., and Nickel, W. (2012) Phosphatidylinositol 4,5-Bisphosphate (PI(4,5)P_2_)-dependent Oligomerization of Fibroblast Growth Factor 2 (FGF2) Triggers the Formation of a Lipidic Membrane Pore Implicated in Unconventional Secretion. J Biol Chem 287, 27659-27669.

3. Zeitler, M., Steringer, J. P., Müller, H. M., Mayer, M. P., and Nickel, W. (2015) HIV-Tat Protein Forms Phosphoinositide-dependent Membrane Pores Implicated in Unconventional Protein Secretion. J Biol Chem 290, 21976-21984.

4. Temmerman, K., Ebert, A. D., Müller, H. M., Sinning, I., Tews, I., and Nickel, W. (2008) A direct role for phosphatidylinositol-4,5-bisphosphate in unconventional secretion of fibroblast growth factor 2. Traffic 9, 1204-1217.

5. Temmerman, K., and Nickel, W. (2009) A novel flow cytometric assay to quantify interactions between proteins and membrane lipids. J Lipid Res.

6. Zacherl, S., La Venuta, G., Müller, H. M., Wegehingel, S., Dimou, E., Sehr, P., Lewis, J. D., Erfle, H., Pepperkok, R., and Nickel, W. (2015) A direct role for ATP1A1 in unconventional secretion of fibroblast growth factor 2. J Biol Chem 290, 3654-3665.

*7.* Müller, H. M., Steringer, J. P., Wegehingel, S., Bleicken, S., Munster, M., Dimou, E., Unger, S., Weidmann, G., Andreas, H., Garcia-Saez, A. J., Wild, K., Sinning, I., and Nickel, W. (2015) Formation of Disulfide Bridges Drives Oligomerization, Membrane Pore Formation and Translocation of Fibroblast Growth Factor 2 to Cell Surfaces. J Biol Chem 290, 8925-8937.

[Editors' note: the author responses to the re-review follow.]

[…] Reviewer #1:

*The major issue raised during review of the original manuscript regarded the confirmatory nature of the manuscript, which confirmed specific requirements that were identified previously using cell based assays. In the revised manuscript, the authors now include several new findings that shed some light onto the mechanism of FGF2 translocation across a membrane. I find the most compelling of these to be the following:*

1) Toroidal translocation intermediate

A lipidic membrane pore with a toroidal architecture containing the FGF2 oligomer in its center is supported by several independent lines of evidence from previous publications and data from our current work. However, the precise molecular mechanism by which membrane-inserted forms of FGF2 oligomers act as translocation intermediates is not fully understood. We have put forward the hypothesis that membrane-inserted FGF2 oligomers are dynamic structures to which FGF2 subunits are constantly added at the inner leaflet concomitant with the removal of subunits at the outer leaflet. In this way, at steady state, membrane-inserted forms of FGF2 have a dynamic oligomeric state in the range of 8 to 12 FGF2 subunits that are continuously remodeled by adding and removing subunits on opposing sides of the membrane. It will be a challenge for future studies to find out whether these assembly/disassembly units are monomers or disulfide-bridged dimers of FGF2, the latter being functional in FGF2 signaling as part of ternary complexes with heparan sulfates and FGF high affinity receptors. Furthermore, a big goal in our lab will be to elucidate the three-dimensional architecture of membrane inserted FGF2 oligomers to fully understand how these complexes are functioning as translocation intermediates in unconventional secretion of FGF2.

As requested by reviewer 1, we have modified the discussion to explain these aspects in more detail.

2) Modifications to Figure 10 and the corresponding legend

We have streamlined Figure 10 with a simplified labelling of its subpanels that makes this figure more self-explanatory. Furthermore, the lipid tracer is now shown for both variant forms of FGF2 which helps to better distinguish the two experimental conditions being compared. The legend has been adapted accordingly.

Reviewer #2:

1) HSQC data from NMR experiments

The full set of HSQC data from the experiments shown in Figure 1 has been added to the source file with all raw data available to the reader.

2) Measurements of PI(4,5)P2 and heparin incorporation into GUVs

Luminal incorporation of heparin into GUVs was monitored by confocal microscopy in control experiments using a fluorescent derivative. Likewise, the presence of PI(4,5)P_2_ in the bilayers of GUVs was analyzed using a recombinant fusion protein consisting of GFP and the PH domain of PLCδ1, a canonical PI(4,5)P_2_ marker. The ‘Materials and methods’ section has been modified accordingly along with the corresponding references.

3) How does the concentration of FGF2 in cells compare with FGF2 concentrations usedin reconstitution experiments?

Though the precise intracellular concentrations of FGF2 in μM has not been reported in the literature, tumor cells massively overexpress FGF2 under pathophysiological conditions which drives an autocrine secretion/signaling loop of FGF2. In our reconstitution experiments, FGF2 is used at low concentrations between 50 and 200 nM (to allow for single molecule measurements) that are likely to be way below the intracellular concentration of FGF2 in tumor cells.

4) Analysis of additional residues in FGF2 with affinity to PI(4,5)P2 as identified by MD simulations

The residues we have identified previously in biochemical experiments (K127, R128 and K133) form a high affinity binding pocket for PI(4,5)P_2_ [Temmerman et al., (2008) Traffic 9:1204-1217]. Consistent with both biochemical experiments and MD simulations, these residues are essential for interactions of FGF2 with PI(4,5)P_2_ containing membranes and are required for FGF2 secretion from cells. The identification of additional residues in FGF2 mediating PI(4,5)P_2_ clustering as identified in MD simulations is a novel finding of the current work with important implications for future studies. However, their detailed analysis in reconstitution experiments and cell-based assays is beyond the scope of the current manuscript.

*Reviewer #3:*

1) Electrostatic aspects of mutually exclusive interactions of FGF2 with various forms of heparin and PI(4,5)P_2_

The electrostatic determinants of FGF2 binding to PI(4,5)P_2_, heparin isaccharides and long-chain heparins are an important aspect with regard to the mechanism of FGF2 membrane translocation we are proposing. However, it is clear from the literature that all of these interactions are not merely based upon the number of electrostatic pairs. Rather, both the orientation of sugar subunits in the heparin backbone and the positioning of sulfate moieties along the ring structure are critical determinants for FGF2 binding. There are in fact examples for different kinds of heparin molecules that are identical in net charge but differ in the positioning of the sulfate groups resulting in strong binding versus a complete lack of FGF2 binding [Li et al., (2014) ACS Chem Biol 9:1712-1717) and papers therein]. Likewise, we have previously demonstrated that FGF2 binding to PI(4,5)P_2_ is much stronger compared to PI(3,4,5)P3 [Temmerman et al., (2008) Traffic 9:1204-1217], pointing at a defined binding pocket for PI(4,5)P_2_ in FGF2 as demonstrated by our NMR experiments. Therefore, hyaluronic acid that lacks sulfate groups even at high concentrations is very unlikely to outcompete PI(4,5)P_2_ with regard to binding to FGF2. As outlined in our manuscript, there are two defined and overlapping binding epitopes in FGF2 for heparin and PI(4,5)P_2_ resulting in mutually exclusive interactions, a key aspect of the translocation model we are proposing.

2) Potential effects of divalent cations on FGF2-heparin interactions

As indicated by reviewer 3, the use of divalent calcium and magnesium ions in biochemical reconstitution experiments employing GUVs is technically challenging. However, in cell-based FGF2 secretion experiments, we found that both calcium (1.8mM) and magnesium ions (0.8 mM) in the extracellular medium does not affect the ability of heparan sulfates to trigger FGF2 membrane translocation and to retain FGF2 on cell surfaces (Zehe et al., 2006, Proc. Natl. Acad. Sci. U.S.A. 103:15479-15484). These observations in intact cells demonstrate that the mechanism we are proposing is functional even in the presence of millimolar concentrations of divalent cations.

3) MD simulations of the interaction between FGF2 and PI(4,5)P_2_.

Reviewer 3 commented on the net charge of PI(4,5)P_2_ and referred to recent work by the Radhakrishnan group [e.g., J Phys Chem B 117, 8322 (2013)]. Their quantum mechanical calculations indicated a net charge of PI(4,5)P_2_ to be –4 around pH 7. This is fully consistent with our work. However, deviations from the net charge of –4 were suggested in the presence of divalent ions. While the effects of divalent salt and its influence on the net charge of PI(4,5)P_2_ could be explored in MD simulations, it would require a computational resource that is larger by a factor of 5–10 compared to the resources used for the present work. In addition, the nature of the results is not expected to change as both –3 and –4 are exceptionally large charges for a lipid head group and will be predominate over any other lipid in terms of electrostatic interactions.

Another aspect raised by reviewer 3 concerned the role of water molecules in the course of FGF2 binding to PI(4,5)P_2_.To address this question, we analyzed our simulation data carefully by computing radial distribution functions, solvent accessible surface areas, number of water molecules around the PI(4,5)P_2_ binding pocket in FGF2 and number of hydrogen bonds between FGF2 binding pocket residues and water molecules. The best overall picture is given by the number of water molecules around the binding pocket region (residues K127, R128, K133). As shown in Figure 15, the number of water molecules is about 95 when FGF2 is not in contact with PI(4,5)P_2_:

Author response image 1.Number of water molecules in the vicinity of the PI(4,5)P_2_ binding pocket in FGF2 in the course of time.The PI(4,5)P_2_ head group binds to the binding pocket at about 500 ns. Here, a water molecule is considered to be in the vicinity of the binding pocket if its distance from the main residues (K127, R128, K133) is 0.6 nm or less.**DOI:**
http://dx.doi.org/10.7554/eLife.28985.036

When PI(4,5)P_2_binds to FGF2 by occupying the binding pocket at about 500 ns, the number of water molecules around the binding pocket region decreases by ~25% to about 70. This is quite expected, since the PI(4,5)P_2_ head group blocks some of the water molecules away from the binding pocket region. Nonetheless, the binding pocket remains very hydrated, i.e. the hydration level does not change in any essential manner.